# σE of *Streptomyces coelicolor* can function both as a direct activator or repressor of transcription

Jiří Pospíšil [1✉], Marek Schwarz [2], Alice Ziková[2], Dragana Vítovská [1], Miluše Hradilová[3], Michal Kolář [3], Alena Křenková[4], Martin Hubálek[4], Libor Krásný [1] & Jiří Vohradský [2✉]

σ factors are considered as positive regulators of gene expression. Here we reveal the opposite, inhibitory role of these proteins. We used a combination of molecular biology methods and computational modeling to analyze the regulatory activity of the extra-cytoplasmic σE factor from *Streptomyces coelicolor*. The direct activator/repressor function of σE was then explored by experimental analysis of selected promoter regions in vivo. Additionally, the σE interactome was defined. Taken together, the results characterize σE, its regulation, regulon, and suggest its direct inhibitory function (as a repressor) in gene expression, a phenomenon that may be common also to other σ factors and organisms.

[1] Laboratory of Microbial Genetics and Gene Expression, Institute of Microbiology of the Czech Academy of Sciences, Vídeňská 1083, 142 20, Prague 4, Czech Republic. [2] Laboratory of Bioinformatics, Institute of Microbiology of the Czech Academy of Sciences, Vídeňská 1083, 142 20, Prague 4, Czech Republic. [3] Laboratory of Genomics and Bioinformatics, Institute of Molecular Genetics of the Czech Academy of Sciences, Vídeňská 1083, 142 20, Prague 4, Czech Republic. [4] Institute of Organic Chemistry and Biochemistry, Czech Academy of Sciences, Flemingovo nam. 542/2, 160 00, Prague 6, Czech Republic. ✉email: jiri.pospisil@biomed.cas.cz; vohr@biomed.cas.cz

Bacteria are highly sophisticated microorganisms that can adapt quickly to changes in their environment. This adaptation is often mediated at the transcriptional level, where the key role is played by multisubunit RNA polymerase (RNAP). The RNAP core is composed of $2\alpha$, $\beta'$, $\beta$, and $\omega$ subunits. To initiate transcription, the RNAP core must bind a sigma ($\sigma$) factor that recognizes specific promoter sequences in DNA. Different bacterial species contain different numbers of $\sigma$ factors. In Gram-positive spore-forming and antibiotic-producing *Streptomyces coelicolor*, 65 genes encoding $\sigma$ factors have been identified[1]. This number is large in comparison with other Gram-positives; *e.g.*, *Bacillus subtilis* contains "only" 19 $\sigma$ factors[2,3].

Usually, one of the $\sigma$ factors is primary and essential for the transcription of most housekeeping genes. HrdB is the primary $\sigma$ factor in *S. coelicolor* and regulates the expression of 2137 genes[4]. The other, alternative $\sigma$ factors regulate transcription of gene subsets that are necessary for survival in different environmental conditions[5]. Extracytoplasmic function (ECF) $\sigma$ factors are a subgroup of small alternative $\sigma$ factors that are important for stress response and during morphological development[6]. ECF $\sigma$ factors are usually regulated by anti-$\sigma$ factors that are embedded in the cell membrane and contain a sensory/signaling domain that responds to extracellular signals (*e.g.*, particular stressors or a lack of nutrients). In response to the signal, the anti-$\sigma$ factors are processed by specific proteases. The respective ECF $\sigma$ factors are then released and become active[7–9].

This study focuses on *S. coelicolor* $\sigma^E$ that belongs to ECF $\sigma$ factors and its regulation is believed to be anti-$\sigma$ factor independent. Previous studies have shown that the amount of $\sigma^E$ in the cell is regulated at the transcriptional level by the CseB/CseC two-component signal transduction system and the CseA protein. After external stress, the CseC sensor kinase transfers phosphate to the transcriptional regulator CseB that turns on the expression of the *sigE* gene, which is part of the *sigE-cseA-cseB-cseC* operon[10,11]. CseA is a lipoprotein localized on the external side of the cytoplasmic membrane and its role in regulation of $\sigma^E$ remains unknown. However, deletion of CseA has been shown to lead to overexpression of the *sigE* gene, suggesting a repressive effect of CseA[12]. $\sigma^E$ is a regulator of the cell envelope stress response as shown by microarray and ChIP-seq analyses after vancomycin treatment. The reported $\sigma^E$ regulon contains 137 genes and more than half of them are involved in cell envelope biosynthesis[13]. No other conditions were investigated.

In this work, we analyzed the cellular level of $\sigma^E$ during various stress conditions and found that EtOH stress strongly induced $\sigma^E$ expression while other stresses had the opposite effect. We then determined the $\sigma^E$ regulon with or without EtOH treatment using ChIP-seq and compared it with the already-known vancomycin-induced regulon. Most importantly, we used our ChIP-seq data together with previously published *S. coelicolor* A3(2) M145 transcriptomic time series microarray data[14] to characterize the $\sigma^E$ regulon. According to kinetic modeling using the transcriptomic time series, we divided the genes of the $\sigma^E$ regulon into three groups (A) genes regulated strictly by $\sigma^E$ (B) genes partially regulated by $\sigma^E$ and/or by other $\sigma$ factors (particularly HrdB) (C) genes with no or constitutive expression. Within these groups, we found that some of the genes could be regulated negatively by $\sigma^E$, suggesting direct or indirect inhibitory effects of $\sigma^E$. The direct effect could be mediated by binding of $\sigma^E$ and/or the RNAP-$\sigma^E$ holoenzyme to its promoter sequence without initiating transcription, functioning as a roadblock and preventing transcription from overlapping/upstream promoters. To test this hypothesis, we analyzed the expression of selected genes in vivo by RT-qPCR in wild type (wt) and $\Delta\sigma^E$ strains. Indeed, as predicted by the modeling, several genes identified by ChIP-seq began to be expressed in the absence of $\sigma^E$. We further tested and validated

this model by mutating $\sigma^E$-dependent promoters that had been predicted to function as positive/negative regulators. In sum, this work illustrates that the presence of peaks identified by ChIP-seq experiments does not necessarily indicate eventual induction of gene expression and that $\sigma$ factors and/or their complexes with the RNAP core can likely act also as barriers (repressors) of gene expression. Finally, interaction partners of $\sigma^E$ were detected, providing a framework for future studies of $\sigma^E$ regulation.

## Results

**$\sigma^E$ expression in different stress conditions**. Tran et al.[13] reported that transcription of $\sigma^E$-dependent genes in *S. coelicolor* is induced upon vancomycin treatment. In our study, we initially analyzed other stress conditions that could potentially affect $\sigma^E$ expression. We used four different stress conditions: osmotic stress by NaCl, oxidative stress by diamide, elevated temperature at 42 °C, and cell envelope stress by EtOH. As a tool, we used a strain where $\sigma^E$ was HA-tagged; the gene was in its natural locus. To verify that the presence of the tag did not affect $\sigma^E$, we tested the sensitivity of the strain to lysozyme treatment that is $\sigma^E$-dependent[15]. Both strains (wt and $\sigma^E$-HA) behaved identically (Supplementary Fig. 1), suggesting that the $\sigma^E$ function was not compromised.

Subsequently, we performed Western blotting experiments with lysates obtained from cells harvested before, and at two time points after the onset of the specific stress. Figure 1 shows that $\sigma^E$ was expressed even under no-stress conditions (time 0). Surprisingly, $\sigma^E$ was rapidly degraded during osmotic, oxidative, and heat stresses. Its expression then increased upon EtOH treatment, and, therefore, we selected these stress conditions for our subsequent experiments.

**ChIP-Seq identifies 193 $\sigma^E$ binding sites**. Next, we identified $\sigma^E$ binding sites across the *S. coelicolor* genome using ChIP-seq. The

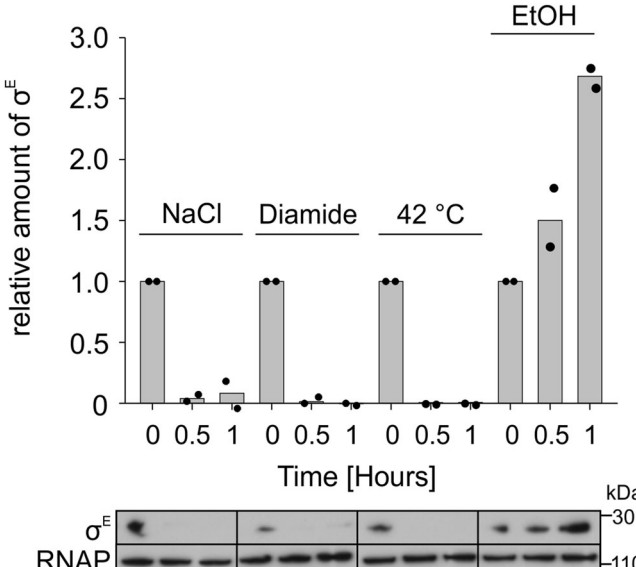

**Fig. 1 $\sigma^E$ cellular levels during various stresses.** Western blot results show cellular levels of $\sigma^E$ in *S. coelicolor* LK3802 during 1 h incubation at 30 °C (or at 42 °C in the case of heat stress) under osmotic (0. 5 M NaCl), oxidative (0.5 mM Diamide), or EtOH (4%) stress. The $\sigma^E$ signal was normalized to the signal of the RNAP core ($\beta$ subunit) and the first time point (0 min) was set as 1. Representative primary data of $\sigma^E$ and RNAP signals are shown below the chart. The experiment was performed in two biological replicates. The bars are averages and the dots represent individual replicas.

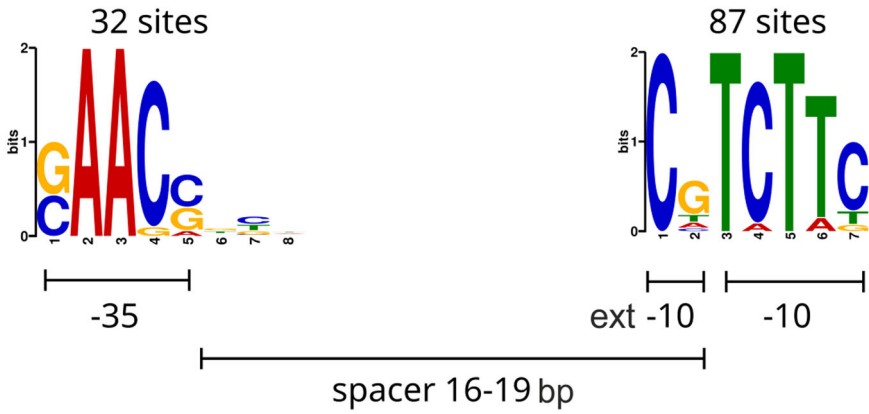

**Fig. 2 Two conserved half-sites discovered with MEME based on ChIP-seq data.** The -10 (containing -10 extended) motif was discovered at 87 sites. The -35 motif was discovered in upstream regions of sequences containing the -10 motif at 32 sites.

experiment was performed in parallel with four biological replicates in the absence/presence of EtOH (see "Methods").

Analysis of the ChIP-seq data identified 193 σ$^E$ binding sites (Supplementary Fig. 2a). Out of the 193 binding sites, 92 (48%) were mapped into the gene coding sequences (intragenic). Five intragenic sites are probably promoters of already annotated small RNAs [(sRNA)][16]. The intragenic sites were not further analyzed due to the lack of transcriptomic data.

Intergenic binding sites were associated with 127 genes including one small RNA (sRNA) potentially controlled by σ$^E$. For the subsequent analysis, only genes immediately following the binding site (i.e., 100 genes) were used as the operon organization was not well defined. Eleven genes were then found exclusively in the EtOH-treated set of experiments (Supplementary Table 3 and Supplementary Data 1). Consistent with the type of stress, functional analysis showed that the only overrepresented functional group was the group of Periplasmic/exported/lipoproteins (hypergeometric test $p$-value < 0.05, 41 genes of 100 total). Highly represented (while not significant) were unknown and non-classified genes (20 and 5 genes, respectively, of a total of 100).

**Promoter analysis reveals -35, -10, and extended -10 elements.** Subsequently, based on the ChIP-seq binding data, we derived the σ$^E$ promoter motif. The sequence logo for the conserved promoter sites is depicted in Fig. 2. The motif half-sites are in good agreement with the recently published motif for *S. coelicolor* σ$^E$ (Supplementary Fig. 2b)[13]. The -35 part of the motif (positions 2–7 in the logo, Fig. 2) surprisingly well matches the -35 binding site of the *E. coli* σ$^E$ consensus GGAACTT[17] while the sequence similarity of the two proteins is only 24% (comparing AA sequences for SCO3356 and AUG17323.1 by Clustal Omega[18]). Most of the identified σ$^E$ binding sites had an 18 bp spacer (59 out of 100 identified), which is consistent with findings by Tran et al.[13]. Motif sites with a spacer length of 19 bp were identified in 21 cases. The remaining σ$^E$ binding sites with spacer lengths of 16 and 17 bp were found in 9 and 11 cases, respectively, consistent with variable spacer lengths of multiple *S. coelicolor* σ factors[19].

**Kinetic modeling detects three divergently regulated gene groups.** Next, the genes associated with σ$^E$ as identified by ChIP-seq were analyzed using the gene expression kinetic database (see "Methods") and clustered according to the correlation between expression profiles. We note that the expression kinetic dataset was obtained from a fermentor study but its high detail (32-time points) allowed us to identify regulatory relationships that were tested later on in this study. Clustering of the expression profiles identified three main

groups: (A) profiles correlated with σ$^E$; (B) profiles correlated with HrdB; (C) flat profiles indicating low within profile variability that could not be correlated with any σ factor (Fig. 3a–c). Additionally, we detected profiles of genes whose kinetics were neither σ$^E$-nor HrdB-dependent but were expressed significantly. The first group of these genes was formed by three genes only (SCO3931, SCO2611, SCO1385), and all three had the σ$^E$-binding motif (SCO2611 had also the HrdB motif) in their promoter regions. The second group consisted of sRNAs, containing six sRNAs (SCR7204, sRNA209, asRNA_SCO6109, ncRNA2772, asRNA_SCO4524, asR-NA_SCO2943). This group could not be modeled due to the absence of kinetic data. The results for Groups A–C are discussed in the following paragraphs.

The genes of Group A, defined as σ$^E$-dependent exclusively (28 genes), are summarized in Supplementary Table 4. Some of them (12 of 28 genes) were also found by Tran et al.[13]. Twelve of them were also identified previously in an HrdB ChIP-seq experiment[4]. Importantly, three of them (SCO4350, SCO5254, SCO5742) were modeled with a negative model parameter w, indicating that σ$^E$ could function as a repressor. SCO4350 represents a putative integrase[20]. SCO5254, also known as SodN, is superoxide dismutase that is highly expressed during the rapid growth of glucose due to elevated levels of reactive oxygen species. Its putative increase after, *e.g.*, diamide stress (due to degradation of σ$^E$), is consistent with the negative regulatory role of σ$^E$. SCO5742 encodes a putative membrane protein with an unknown function. Overall, the functional analysis of Group A showed an overrepresentation of genes encoding membrane proteins (Sanger group Periplasmic/exported/lipoproteins—16 out of 28 genes (57%), which is 3.4 times more than the incidence of these proteins in the genome (16.8%)).

Group B contained HrdB-dependent genes, i.e., those genes that were identified in the σ$^E$ ChIP-seq experiment but could be modeled exclusively with the HrdB expression profile (Supplementary Table 5), suggesting that HrdB is the dominant regulator. This group comprised 16 genes, and except for three (SCO4468, SCO4244, and SCO3609), all possess a HrdB binding motif in their promoter region. Nine of them were also identified in the ChIP-seq experiment reported in Smidova et al.[4]. Functional analysis then revealed enrichment of genes encoding membrane proteins (6 of 16; 38%).

To assess the potential role of HrdB in the expression of genes identified by σ$^E$ ChIP, we used the previously published HrdB ChIP-seq data[4] and performed a search for peaks present in both σ$^E$ and the HrdB ChIP-seq experiments. This search yielded 64 peaks common for both σ factors. This necessitated further analysis addressing the involvement of HrdB in the regulation of

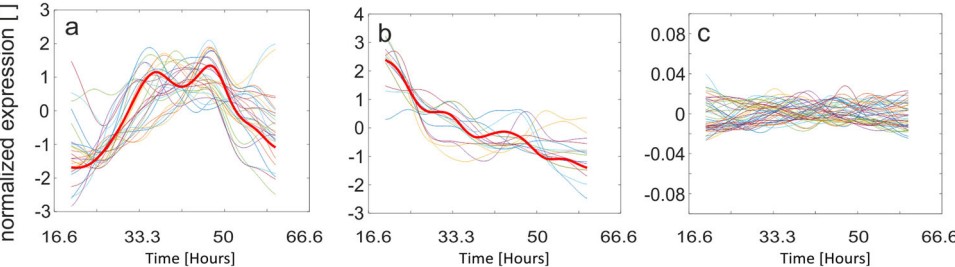

**Fig. 3 Normalized expression profiles of gene clusters identified by ChIP-seq. a** Profiles correlated with σ^E (red thick line), **b** Profiles correlated with HrdB (red thick line), **c** Cluster with flat profiles (note the scale which is two orders lower than that of the panels a, b, indicating low within profile variability).

the σ^E regulon. The expression profiles of groups A and B were modeled using σ^E and/or HrdB as regulators and the results were combined with the peak search results and promoter sequence analysis (Supplementary Data 1 and/or Supplementary Tables 4–6), indicating that HrdB and σ^E can together control some genes.

Group C contained genes with flat or no detectable expression profiles. Although the genes of this group contain σ^E and/or σ^E and HrdB binding motifs, their expression profiles could not be correlated with σ^E or HrdB. This, however, did not exclude the possibility of their regulation by σ^E and HrdB in combination with other, unknown regulators. Functional analysis showed, as in the previous cases, a prevalence of the genes encoding membrane proteins (17 of 47). From the functional point of view, the most abundant genes found in the three groups defined above were genes encoding "periplasmatic/exported/lipoproteins".

The three groups, however, differ in the percentage ratios of the number of genes in this functional group with respect to all genes, when the percentage ratio was highest for Group A (σ^E-dependent genes, Supplementary Table 4, 57%) and lowest for genes in Group C (Supplementary Table 6; 36%).

To conclude this part, the analyses of Groups A–C implied that in some cases, σ^E, although bound, does not initiate transcription of the downstream genes. Moreover, it may even block the expression of genes positively regulated by HrdB. The regulation may even be more complicated with the involvement of other so far unidentified regulators. To provide insight into these alternatives, we designed a set of experiments where we tested the expression of selected genes (in the absence/presence of EtOH) in wt and Δσ^E strains by RT-qPCR.

**Identification of positively and negatively σ^E-regulated genes.** Based on the modeling of expression described in the previous section and to gain more insights into the regulatory role of σ^E, we performed a panel of RT-qPCR experiments with selected genes from groups A–C.

As Group A consists of positively and negatively regulated genes, we selected three positively (SCO3396, SCO7657-hyaS, SCO6722-ssgD,) and one negatively (SCO4350) regulated gene for subsequent RT-qPCR validation. SCO3396 and hyaS were selected based on previous work where it was shown that these genes contain a single σ^E-dependent promoter[20]. SCO3396 encodes a cell envelope-associated enzyme (putative esterase). HyaS, an amine oxidase, is a copper-binding enzyme containing a twin-arginine motif suggesting its secretion via the Tat-transport system. In *Streptomyces lividans*, HyaS (99% homolog of *S. coelicolor* HyaS) affects hyphae aggregation during normal growth, probably due to induction of cross-links with other hyphae-associated protein(s) or other compound(s)[21,22]. The ssgD gene was not previously reported to belong to the σ^E regulon. The function of SsgD is related to peptidoglycan synthesis along the lateral cell wall and its expression increases

after treatment with cell wall synthesis-targeting antibiotics (vancomycin, bacitracin, or moenomycin)[23–25]. As already mentioned, negatively regulated SCO4350 encodes a putative integrase. The RT-qPCR results confirmed that the expression of SCO3396, hyaS, and ssgD was positively controlled by σ^E. Interestingly, expression of hyaS was already high in the absence of EtOH, and EtOH addition did not lead to its further increase. ssgD and SCO3396 were upregulated after EtOH treatment in a σ^E-dependent manner. Importantly, SCO4350 was negatively regulated by σ^E in both conditions (Fig. 4). Its inhibition in the presence of EtOH in Δσ^E is due to an unknown mechanism.

From Group B (HrdB-dependent, Fig. 5), we tested SCO1595-pheS, SCO5393, SCO5535-accB, and SCO4921-accA2 as genes likely positively regulated by HrdB. Here, the selection was made based on the presence of predicted HrdB binding sequences upstream of the genes. Except for SCO5393, all the other tested genes were previously identified as members of HrdB regulon[4]. Nevertheless, the presence of σ^E binding sites suggested a possible role also for this σ factor. It was already shown that accB is a member of σ^E regulon[13]. The pheS gene is expressed in one operon (sco1595-sco1594) together with pheT and both encode subunits of putative phenylalanyl-tRNA synthetase. SCO5393 is an ATP-binding subunit of a putative ABC transporter. The accB gene is the first gene of the sco5535-sco5536 operon and together with accA2 encodes the multienzyme complex converting acetyl-CoA to malonyl-CoA[26].

The results showed that pheS was expressed similarly in both genetic backgrounds, suggesting a negligible effect of σ^E on its expression. In the presence of EtOH, expression decreased. Expression of SCO5393 was decreased in Δσ^E relative to wt under normal conditions, suggesting that σ^E may contribute to its expression. Moreover, SCO5393 exhibited lower expression under EtOH treatment in the wt strain. Expression of accB and accA2 genes was negatively affected by σ^E in both conditions. Therefore, we created a model that combined HrdB as an activator and σ^E as a repressor, resulting in more reliable kinetics of their expression (Supplementary Fig. 3). Both genes are known members of the HrdB regulon and HrdB promoter motifs were bioinformatically identified. Consistent with the repressor role of σ^E, the HrdB motifs are overlapped with the σ^E motif (Fig. 5)

Group C consists of genes with flat or weak expression, therefore we did not expect their significant regulation by σ^E (at least in normal conditions). Here the genes were selected randomly as they are not expressed in normal conditions and most of their products are not characterized (SCO3796, SCO5213, SCO5981, and SCO3824). SCO4615 is an integrase that is part of the genomic island that was apparently created by the insertion of the pSLP2 vector from *S. lividans*[27,28]. SCO6532 is part of the sco6532-sco6535 operon encoding four subunits of Nar1, a spore-specific nitrate reductase[29,30]. As shown in Fig. 6a, SCO3796, SCO5213, and SCO4615 were not affected by the absence of σ^E. SCO3796 and SCO5213 were upregulated in EtOH

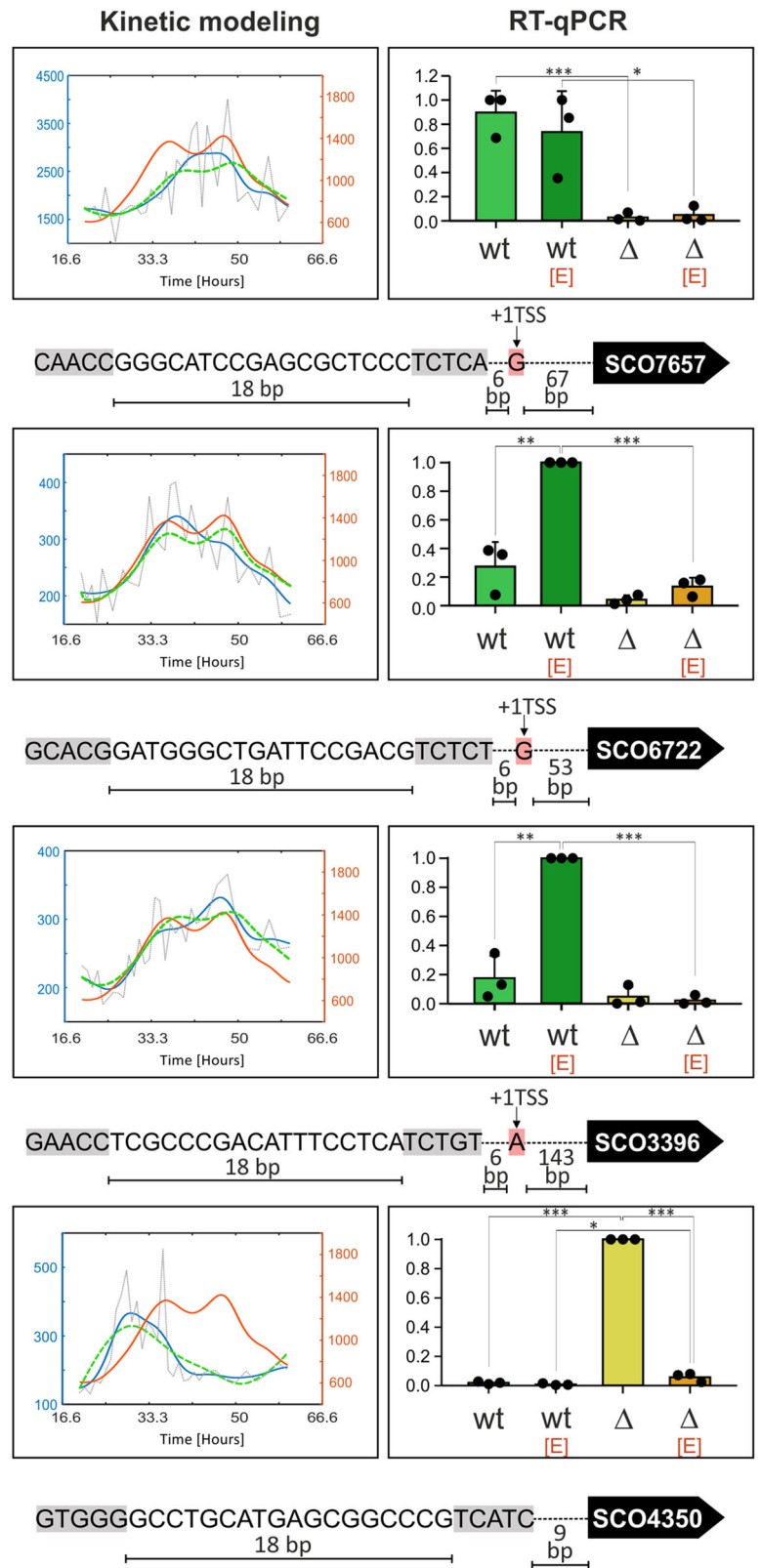

stress in wt and $\Delta\sigma^E$ strains respectively, but without $\sigma^E$ contribution. Surprisingly, group C also contains genes with low expression in wt but high in the $\Delta\sigma^E$ strain (SCO5981, SCO3824, and SCO6532). SCO5981 was similarly repressed by $\sigma^E$ in both conditions but its expression was not affected by EtOH stress in either genetic background. Interestingly, SCO3824 and SCO6532 were repressed by $\sigma^E$ only under EtOH stress. SCO6532

was induced by EtOH. However, this induction was clearly inhibited by $\sigma^E$ (Fig. 6b).

Taken together, RT-qPCR results showed that binding of $\sigma^E$ to the promoter region (ChIP-seq analysis) does not necessarily mean positive regulation of downstream genes. Genes of the $\sigma^E$ regulon could be regulated also negatively by this $\sigma$ factor. Consistently, this repression is disrupted in $\Delta\sigma^E$. Thus, in this

**Fig. 4 Experimental validation of selected genes from Group 1 ($\sigma^E$-dependent).** Expression of genes predicted to be positively (SCO7657-*hyaS*, SCO6722-*ssgD*, SCO3396) or negatively (SCO4350) regulated by $\sigma^E$. The charts on the left represent the results of the computational modeling based on Nieselt et al.[14]. The orange line (and the right orange vertical axis) represents the relative expression of the regulator ($\sigma^E$), the blue line (and the left blue vertical axis) relative expression of the regulated gene, the gray dotted line primary data of regulated gene expression and the green line a regulation model. The charts on the right show the relative expression (vertical axis) of specific genes according to RT-qPCR (for details see "Methods"). Each gene was analyzed in the wt (LK3801) and $\Delta\sigma^E$ [$\Delta$] (LK3804) strains that were or were not treated with EtOH [E]. The experiment was performed in three biological replicates. Bars are averages and error bars represent StDev. Dots represent individual replicates. Two-tailed *t*-test *p*-values that are less than 0.001, 0.01, and 0.05 are marked as \*\*\*, \*\*, and \*, respectively. Selected consensus and other promoter features are shown below the charts. The gray highlights show -10 and -35 $\sigma^E$ promoter regions. The pink highlights show +1 transcription start sites (+1TSS, if identified in ref. [16]) downstream of the promoter.

negative regulation, $\sigma^E$ either functions (i) directly as a repressor, likely in the form of the RNAP-$\sigma^E$ that binds to its recognition sequence and blocks the binding of a positive regulator or represents a barrier to transcription, or (ii) indirectly as a regulator of another repressor.

**$\sigma^E$ functions as a direct negative regulator.** To distinguish between the direct and indirect roles of $\sigma^E$, we selected two genes from Group A that were significantly positively (SCO7657-HyaS) or negatively (SCO4350) regulated by $\sigma^E$. We mutated -10 elements in their $\sigma^E$ promoter regions, abolishing the binding of $\sigma^E$. Mutated (or non-mutated) promoters, containing upstream regions, were cloned into the pBPSA1 vector, driving the expression of a marker gene encoding indigoidine synthetase[31]. The constructs were ectopically integrated into the wt strain and relative promoter activities were measured spectrophotometrically (see "Methods"). Consistent with RT-qPCR results, mutation of the $P_{SCO7657}$ -10 region led to downregulation of its expression (Fig. 7a). Importantly, the same type of mutation in the $P_{SCO4350}$ promoter -10 region caused transcription upregulation (Fig. 7b), consistent with a role of $\sigma^E$ as a direct repressor.

Based on RT-qPCR and indigoidine quantification experiments we used $\sigma^E$ binding sites of three positive (POS; *hyaS*, *ssgD*, SCO3396) and four negative (NEG; SCO4350, *accB*, *accA2*, SCO5981) $\sigma^E$ regulated genes. Using these sequences we created their promoter motif logos (Fig. 8a, b, upper panel). The POS and NEG logos markedly differed. In the -35 region, the NEG logo differed in most positions as its sequence conservation was low. In the -10 region, logos were more similar but their -6 positions completely differed. We stress, though, that the number of sequences that were available for their creation was small.

**Interacting partners of $\sigma^E$.** Finally, we wanted to characterize binding partners of $\sigma^E$ to provide insights into other factors potentially regulating $\sigma^E$-dependent expression. Therefore, we performed a pull-down assay of HA-tagged $\sigma^E$ with or without EtOH stress. In both conditions, $\sigma^E$ was pulled down with RNAP core subunits ($\alpha\beta\beta'\omega$) and HupA, a DNA binding protein[32]. In non-stressed conditions exclusively, we identified a putative anti-$\sigma$ factor ApgA with unknown target, transcriptional regulators BldC and CarD, termination factor Rho, and two putative nucleic acid binding proteins (SCO5570 and SCO0174) (Fig. 9a). On the other hand, nucleoid-binding proteins Lsr2 and sIHF, and transcriptional regulator NdgR, were detected only in stress conditions (Fig. 9b).

## Discussion

This study characterizes the expression of $\sigma^E$ with respect to different types of stress, identifies NaCl, diamide, and heat as stressors that lead to degradation of $\sigma^E$, and EtOH as an inducer of its expression. Furthermore, binding sites of $\sigma^E$ were identified across the *S. coelicolor* genome and correlated with expression kinetics of respective genes and binding sites for $\sigma^E$ and the main

$\sigma$ factor, HrdB. Several groups of genes were identified with respect to regulation by $\sigma^E$ and HrdB. Most notably, the negative regulatory role of $\sigma^E$ was detected and $\sigma^E$ was subsequently shown to likely function as a direct repressor of gene expression, sitting on the promoter and blocking transcription. Finally, we identified binding partners of $\sigma^E$ in non-stressed and EtOH-stressed conditions.

A previous study reported increased expression of *S. coelicolor* $\sigma^E$ after exposure to cell envelope stress caused by cell wall-targeting antibiotics (vancomycin, moenomycin A, and bacitracin)[24,33]. This was subsequently verified by Tran et al. who defined the $\sigma^E$ regulon under vancomycin stress by ChIP-seq analysis and microarrays. The $\sigma^E$ vancomycin regulon contains 137 genes involved in cell wall synthesis, membrane integration, and sporulation[13]. In this work, we detected induction of $\sigma^E$ after EtOH treatment, reminiscent of a study where upregulation of a $\sigma^E$ family $\sigma$ factors (*vnz_RS00060*, *vnz_RS15950*, *vnz_RS18620*, *vnz_RS22785*, *vnz_RS23885*) in *Streptomyces venezuelae* was observed after EtOH treatment[34]. Consistently, it was shown that mycelia of *Streptomyces albus* became thicker and more branched after EtOH treatment suggesting changes in cell envelope and possibly involvement of $\sigma^E$ regulon. These changes were coupled with increases in intracellular reactive oxygen species and changes in the ratio of un/saturated fatty acids in the cell membrane which also could improve fluidity of the cytoplasmic membrane[35].

In our work, $\sigma^E$ was surprisingly degraded during osmotic, oxidative, and heat stresses, suggesting that it is not required in these stress responses. Depletion of a $\sigma$ factor as a consequence of a changing environment was previously reported for *Streptococcus mutans* where $\sigma^X$ (positive regulator of natural competence) is rapidly degraded during escape from the competent state[36].

Using ChIP-seq, we detected 193 $\sigma^E$ binding sites across the genome. Almost half of them (92 out of 193) were mapped into gene coding sequences. Intragenic binding of $\sigma$ factors was shown previously for $\sigma^{37}$, $\sigma^{38}$, and FliA in *Escherichia coli, Salmonella typhimurium*, and *Yersinia pseudotuberculosis*. These binding sites could function as promoters for asRNA transcription but mostly they are transcriptionally inactive while still binding RNAP$\sigma$ holoenzymes; they are activated only under specific conditions[39–43]. Intragenic promoter-like sequences can act also as regions for re-association of RNAP with $\sigma$ factor, resulting in transcriptional pausing[44]. However, in the present work, we did not perform an in-depth analysis of the identified intragenic binding sites due to the absence of kinetic data. The intergenic binding sites were then associated with 127 genes (some of them organized in operons) and one sRNA. These sites could be correlated with kinetic data and are further discussed.

Eleven $\sigma^E$ binding sites were exclusively found in EtOH-stressed cells. These 11 targets represent cell envelope-associated proteins (SCO1550, SCO1963, SCO3046, SCO5511), enzymes threonyl tRNA synthetase-SCO3778, hydrolase-SCO5943 and transcriptional regulators such as multiple antibiotic resistance regulator (SCO3133-*marR*), undefined regulator SCO3423, and $\sigma$

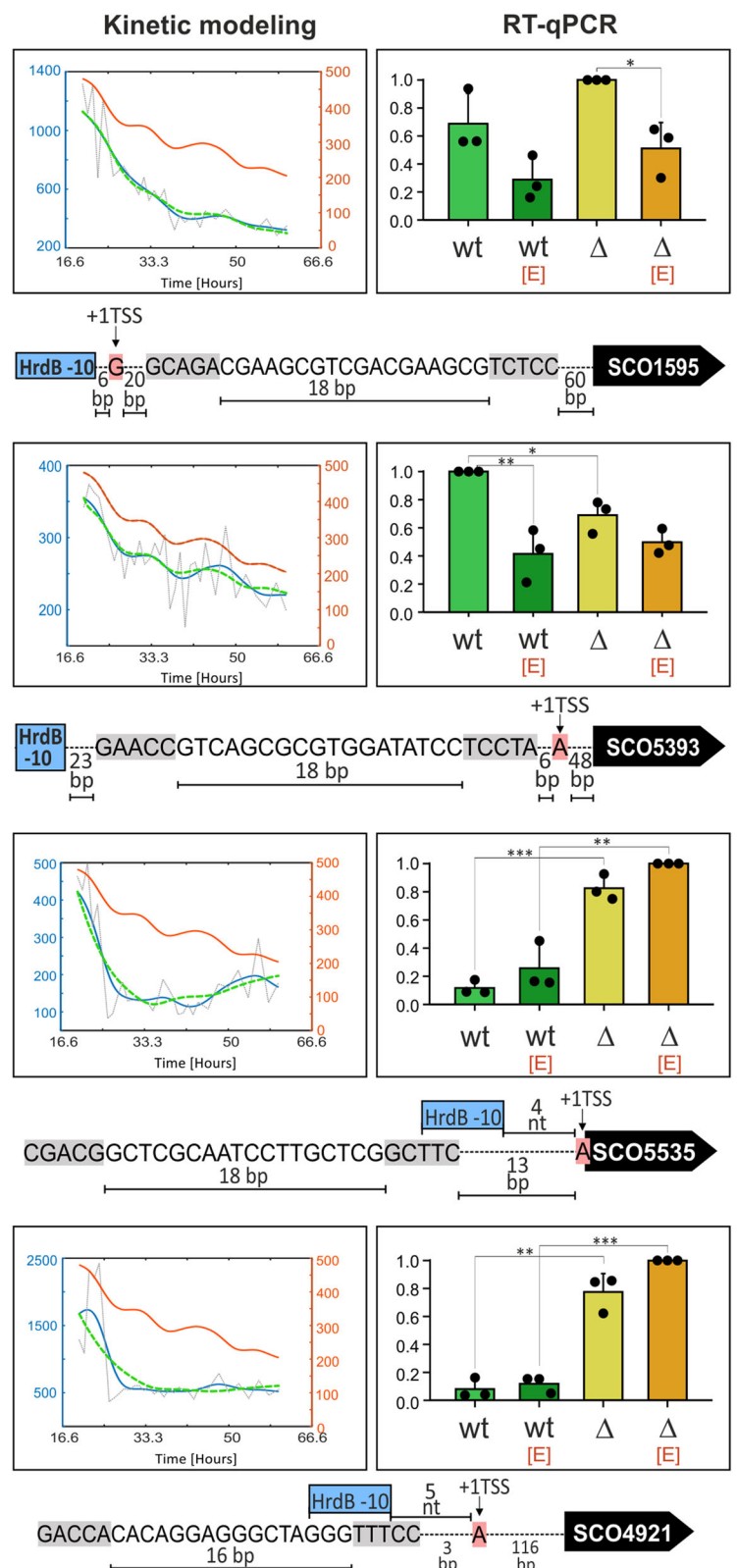

**Fig. 5 Experimental validation of selected genes from Group 2 (HrdB-dependent).** Expression of genes that are predicted to be regulated by HrdB instead of σ[E] (SCO1595-*pheS*, SCO5393, SCO5535-*accB,* and SCO4921-*accA2*). The Figure description is the same as in Fig. 4. The blue rectangles show the -10 region of HrdB-dependent promoters (if identified in ref. [4]).

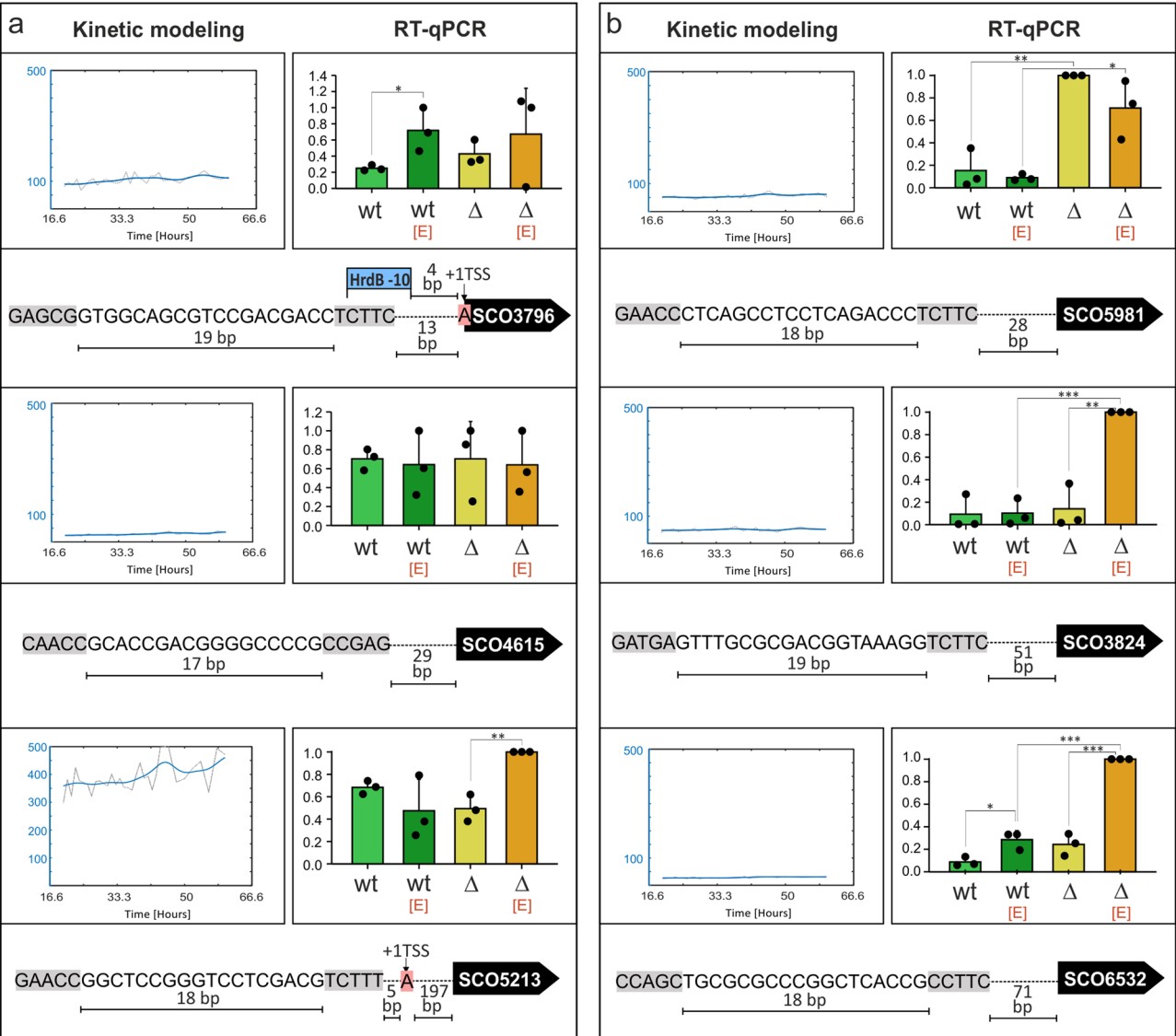

**Fig. 6 Experimental validation of selected genes from Group 3 (Flat).** Expression of genes with flat kinetic profiles. **a** Genes whose expression was not affected by σ[E]. **b** Genes repressed by σ[E]. The figure description is the same as in Fig. 4.

factor Q (SCO4908). Consistent with the type of stress, σ[Q] has been described as a σ factor whose expression is increased in response to cell envelope stress[24]. A comparison of the σ[E] regulon as defined in this current study (132 genes, including 5 genes for asRNA having intragenic promoters) with the already published vancomycin-induced regulon (137 genes), then revealed 57 genes present in both data sets, expanding the known σ[E] regulon to 212 genes (see Supplementary Data 4). Surprisingly, none of the EtOH-induced genes were found in the known vancomycin regulon despite the fact that σ[Q] (SCO4908) has been shown as a vancomycin-induced gene[24]. This indicates that the σ[E] regulon is expressed in dependence on the type of stress, and various types of stress induce different subsets of the regulon.

The σ[E] consensus binding motif determined in our study is highly similar to the previously published motif[13]. The main differences between the two motifs are in the -10 region where, in our logo, the motif is more complex and consistent with the extended -10 motif. Interestingly, the σ[E] consensus sequence of *S. coelicolor* is highly similar to the -35 element and partially similar to the -10 element of the σ[E] consensus which was found in *E. coli* and other ECF σ factors in *Mycobacterium tuberculosis* (σ[E], σ[H]) and *Bacillus subtilis* (σ[X], σ[W])[17,45]. Consistently, in these

organisms, σ[E] homologs also react to cell envelope stress and are important for survival in hostile environments[17,46].

The identified σ[E] regulon was subjected to kinetic analysis. It allowed us to divide the σ[E] regulon into three main groups. A similar sorting had been also carried out previously with genes from the vancomycin regulon after S1 nuclease mapping[13]. According to the ChIP-seq and kinetic modeling, we were able to identify 28 genes as strictly dependent on σ[E]. This included genes that are regulated by σ[E] positively or negatively. The rest of the genes from the σ[E] regulon were classified as HrdB-dependent or as genes with flat or low expression (*i.e.*, not controlled by σ[E] under normal conditions of growth). As apparent from the kinetic modeling data (Fig. 3, the first 30 h), σ[E] and HrdB mRNA levels may behave reciprocally, consistent with the model where changing levels of the sigma factors competing for binding to a limited pool of RNAP core enzymes provide a mechanism for cross-talk between genes or gene classes, ultimately affecting their expression[47].

Using RT-qPCR, we analyzed the expression of several selected genes, and the results of kinetic modeling were confirmed in the vast majority. Importantly, we showed that some of those genes (SCO5981, *accB*, *accA2*, SCO4350) were upregulated in the

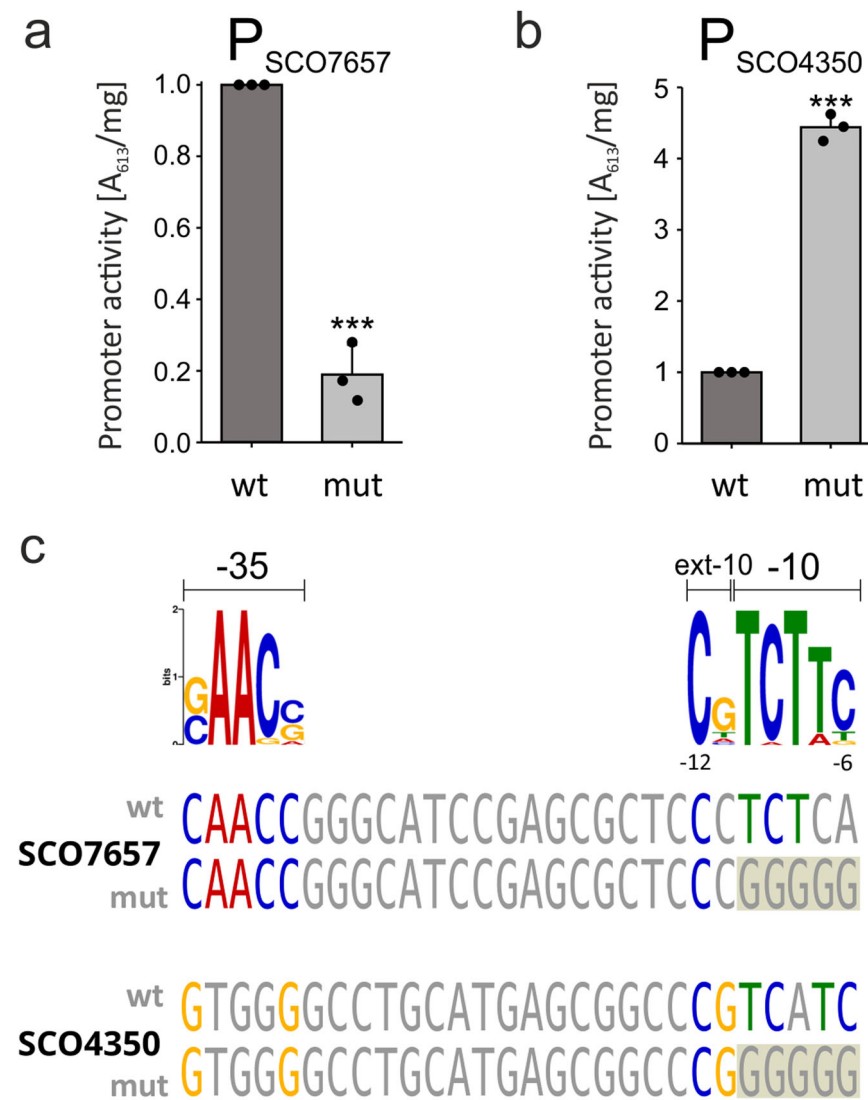

**Fig. 7 Mutations of -10 promoter regions of positively (POS) or negatively (NEG) regulated σ$^E$ genes. a** Activities of wt (LK3626 strain) and mutated (LK3630 strain) SCO7657 promoters in the wt strain (LK3801) background. **b** Activities of wt (LK3633 strain) and mutated (LK3635 strain) SCO4350 promoters in the wt strain (LK3801) background. For both charts: Bars are averages, spots represent individual replicates, error bars represent StDev and the two-tailed *p*-value is less than 0.0001 (***). **c** Comparison of -35 and -10 regions in SCO7657 and SCO4350 promoter sequences. The upper panel represents discovered motif half-sites (as in Fig. 2). The lower panel represents a comparison of selected promoters and their mutated variants (the mutated region is highlighted with tan color) with the primary logo (identical bases are colored the same).

absence of σ$^E$, confirming its negative effect on their expression. SCO5981 is a hypothetical protein with an unknown function. The *accB* and *accA2* genes encode enzymes that are involved in the conversion of acetyl-CoA to malonyl-CoA which are precursors for the synthesis of blue-pigmented antibiotic actinorhodin and red-pigmented bioactive compound undecylprodigiosin with anti-cancer, anti-fungal and anti-malarical properties[48,49]. Previously, it was shown that an actinorhodin-deficient strain is far more sensitive to diamide stress, suggesting that actinorhodin could have anti-oxidative properties[50]. As we showed in this study, diamide stress leads to rapid degradation of σ$^E$ which could result in the expression of actinorhodin pathway genes *accB* and *accA2*. Consistently, it was shown that actinorhodin production was induced in the Δ*sigE* strain, showing a negative effect of σ$^E$[15]. The red-pigmented undecylprodigiosin stimulates programmed cell death in *S. coelicolor* and, therefore, σ$^E$-mediated repression of undecylprodigiosin production (in non-stressed conditions) could protect the cell from accumulating this harmful metabolite[51]. Last but not

least, SCO4350 is another example of a gene that is repressed by σ$^E$. It is a putative integrase, containing the SAM domain that is able to cleave DNA. Hypothetically, repression of SCO4350 may prevent its expression during non-stressed conditions, guarding chromosome integrity[20].

The RT-qPCR experiments themselves still allowed for the possibility that the σ$^E$-dependent repression was mediated by some σ$^E$-dependent proteins and not by σ$^E$ itself. Mutation of the σ$^E$ binding -10 promoter region of SCO4350 then revealed that binding of σ$^E$ to the promoter was essential for the negative regulation, strongly suggesting that σ$^E$ itself can be the repressor. This finding reveals a new, so far unappreciated, inhibitory role of σ factors. It is consistent with previously published data where the presence of a particular σ factor binding to DNA did not correlate with positive regulation of the downstream gene[4,52–54], but experimental validation of the phenomenon was lacking. We stress, however, that not in all cases the negative regulatory role of a σ factor is direct. To distinguish between direct and indirect effects, individual genes/promoters need to be analyzed

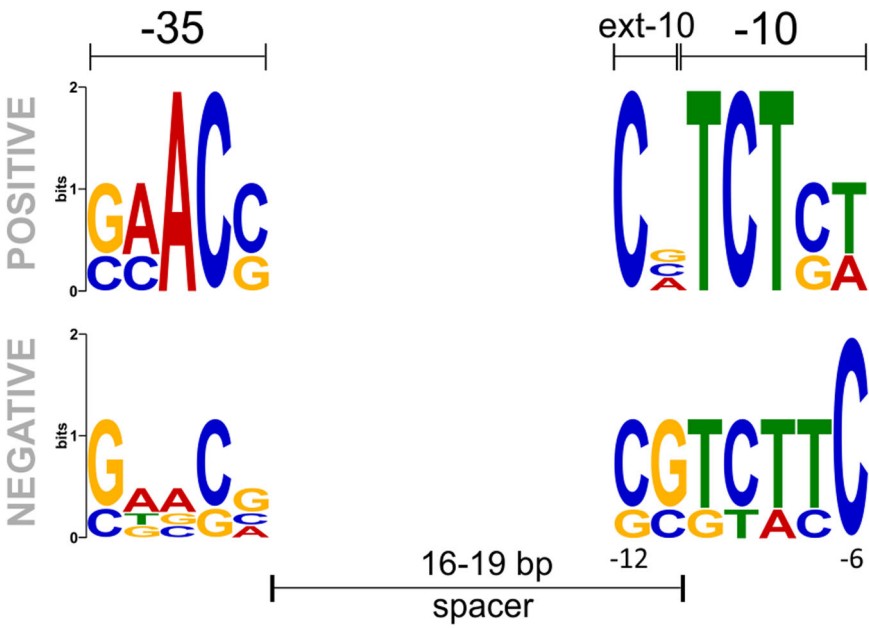

**Fig. 8 Motifs of positively and negatively regulated promoters.** The motif of the positively regulated promoter (upper panel) was created using predicted promoter sequences of 3 positively (POS) regulated genes (*hyaS*, *ssgD*, SCO3396). The bottom panel represents a motif of the negatively regulated promoter, created using predicted promoter sequences of 4 negatively (NEG) regulated genes (SCO4350, *accB*, *accA2*, SCO5981).

experimentally. Likewise, future in vitro studies will be needed to unravel the mechanistic details of this phenomenon.

How does σ$^E$ then function as a direct repressor? Unlike the housekeeping σ factors, σ$^E$ lacks autoinhibitory domain 1.1 preventing its binding to the promoter[55]. Therefore, σ$^E$, in theory, might bind to the promoter without RNAP but this is less likely as previous attempts showed only weak interaction of the σ$^{38}$ family factors with DNA[37]. The other possibility is that σ$^E$ binds to the promoter-like sequence in the form of the RNAP holoenzyme. In either case, deletion of σ$^E$ leads to derepression, and transcription can be initiated by another holoenzyme from an upstream/overlapping promoter recognized by another positive regulator (σ factor). Therefore, we bioinformatically searched for possible positive regulators of SCO4350. We modeled the kinetics of SCO4350 with all the identified σ factors (that could act as stimulators of transcription) together with σ$^E$ (acting as a repressor). The best results were obtained for the combination of σ$^E$ (repressor) and σ$^R$ (SCO5216, activator) or BldN (SCO3323, activator) (Supplementary Fig. 4a). σ$^R$ and BldN are ECF σ factors reacting to disulfide stress or maintaining the formation of aerial mycelium respectively[56,57]. Consistently, bioinformatic search identified putative BldN and σ$^R$ binding sites upstream of, or overlapping the σ$^E$-dependent promoter (Supplementary Fig. 4b, c). A future study will be needed to unravel the details of the positive regulation of the SCO4350 gene.

A similar phenomenon was described in *Y. pseudotuberculosis* where the RNAP-σ$^{37}$ holoenzyme binds to its cognate sequence and represses transcription of downstream gene[39]. In *E. coli*, the activity of some of the promoters recognized by the RNAP-σ$^{37}$ holoenzyme increased in the absence of σ$^{37}$. There, the σ$^{37}$-dependent promoter overlaps with the promoter recognized by other RNAP holoenzymes and represses transcription of the downstream gene. Nitrogen depletion stress then induces expression of enhancer protein NtrC that binds to DNA and RNAP-σ$^{37}$ holoenzyme which initiates transcription from silenced promoters[58]. It is possible that a similar mechanism of activation could exist also for promoters that were found to be repressed by σ$^E$ in our study, and this activation would occur under not yet identified conditions. Nevertheless, the architecture

of σ$^{37}$-dependent promoters (-24 and -12 regions) differs from that recognized by σ$^{38}$ family (-35 and -10 regions) factors, and transcription driven by σ$^{37}$ requires an enhancer protein and ATP hydrolysis[40].

To sum up, this part of the study reveals a new type of transcriptional regulation by σ factors when they act as repressors. In other words, binding to a promoter region itself does not automatically guarantee transcription of the downstream gene but depends on environmental conditions where other factors and regulatory circuits affect the outcome. A model of regulation of σ$^E$ expression is depicted in Fig. 10.

Furthermore, using mass spectrometry, we identified several proteins enriched in σ$^E$ immunoprecipitations in non-stressed or stressed (EtOH) conditions. Besides subunits of RNAP, we identified in non-stressed conditions transcriptional regulator BldC, binding direct repeats upstream of promoters and regulating 369 genes including the *sigE* gene. After a comparison of σ$^E$ and BldC regulons, we found 13 genes that are likely regulated by both of them (SCO3194, SCO5605, SCO4934, SCO1749, SCO1755, SCO3396, SCO3030, SCO4439, SCO2970, SCO3971, SCO2897, SCO3034, SCO3580)[59].

Another transcriptional regulator was CarD. In Streptomyces, CarD is poorly characterized but it is intensively studied in other Actinobacteria such as *Mycobacterium tuberculosis*, where CarD is essential (*Sco* CarD vs. *Mtb* CarD = 85% homology)[59,60]. CarD stabilizes the RNAP holoenzyme/DNA complex and activates or represses gene expression in dependence on promoter sequence. To date, CarD was identified only in holoenzymes containing the main σ factor. Hence, the association of CarD with σ$^E$ (likely in the RNAP holoenzyme form) is the first time when CarD binds to an RNAP holoenzyme containing an ECF σ factor.

Finally, we identified ApgA to associate with σ$^E$ in non-stressed conditions. ApgA was previously suggested to be an anti-σ factor but the identity of the σ factor had been unknown[61,62]. Consistent with the anti-σ factor function of ApgA for σ$^E$ is their association in non-stressed conditions and disappearance of ApgA after stress, as well as the presence of ApgA in the σ$^E$ regulon as reported by[13]. The ApgA encoding gene is co-transcribed with the gene coding for BldG, its potential anti-anti-σ factor[61,62].

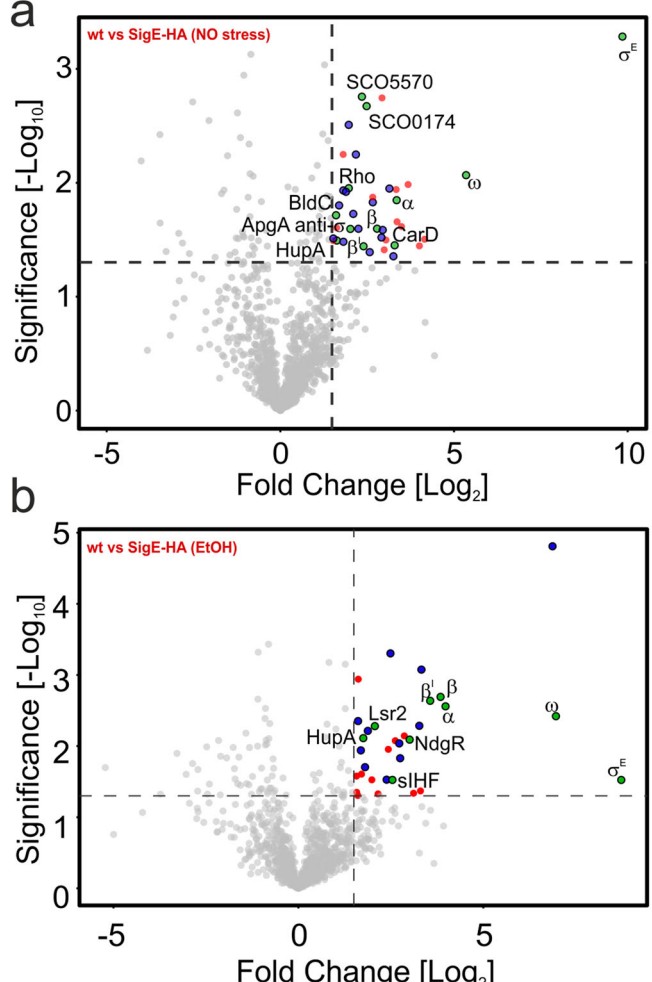

**Fig. 9 Mass spectrometry analysis of σ$^E$ binding partners. a** Pulldown of σ$^E$-HA (LK3802 strain) in non-stressed conditions (equal to time 0 min in western blot experiment in Fig. 1). The same pull-down experiment was performed with the wt strain (LK3801) as a negative control and obtained data were used as a background (gray dots). Thresholds (dashed lines) were set-up as 1.3 and 1.5 for fold-change (x-axis) and Significance (y-axis), respectively. Significantly enriched ribosomal proteins (red dots), unknown proteins (blue dots), and transcription or DNA binding proteins (green dots) are highlighted. **b** Pulldown of σ$^E$-HA (LK3802 strain) in EtOH-stressed conditions (equal to time 60 min in western blot experiment). The same pull-down experiment was performed with the wt (LK3801) strain as a negative control and obtained data were used as a background (gray dots). Proteins are highlighted as in chart **a**.

Future studies will be necessary to explore the effects of the identified σ$^E$ interacting partners on σ$^E$-dependent expression.

## Methods

**Strains and growth conditions**. The strains and plasmids used in this study are listed in Supplementary Table 1. The Streptomyces strains were generally grown in the same way for all subsequent analyses (Western blot, ChIP-seq, RT-qPCR)[4,14]. Briefly, spores were prepared by harvesting them from agar plates grown for 10 days. Subsequently, spore germination was conducted at 30 °C for 5 h by inoculation of spores ($2.5 \times 10^9$/ml CFU) into 2YT medium (bacto tryptone, 16 g/l; bacto yeast extract, 10 g/l; NaCl, 5 g/l). After germination, cultures were washed by 5 ml of ion-free water and inoculated into Na-glutamate medium (Na-glutamate, 61 g/l; glucose monohydrate, 44 g/l; MgSO₄, 2.0 mM; Na₂HPO₄, 2.3 mM;

KH₂PO₄, 2.3 mM) supplemented with 8 ml/l of trace element solution (ZnSO₄×7H₂O, 0.1 g/l; FeSO₄ × 7H₂O, 0.1 g/l; MnCl₂×4H₂O, 0.1 g/l, CaCl₂×6H₂O, 0.1 g/l; NaCl, 0.1 g/l) and 5.6 ml/l of TMS1 (FeSO₄×7H₂O, 5 g/l; CuSO₄×5H₂O, 390 mg/l; ZnSO₄×7H₂O, 440 mg/l; MnSO₄×H₂O, 150 mg/l; Na₂MoO₄×2H₂O, 10 mg/l; CoCl₂ × 6 H₂O, 20 mg/l; HCl, 50 ml/l)[63]. Bacteria were shaken for 23 h (corresponding to exponential phase[64] at 30 °C, 250 rpm, and pH 7 was maintained during the growth. After 23 h, the culture was harvested and processed specifically for Western blot, ChIP-seq, or RT-qPCR (described separately in the following paragraphs).

**Construction of epitope-tagged mutant strain**. The σ$^E$ protein was tagged on its C-terminus by insertion of the HA-tag sequence into the genome (TAC CCA TAC GAC GTC CCA GAC TAC GCT). A gene cassette containing FLP recombinase recognition target (FRT) flanking regions, apramycin resistance, and oriT was amplified by PCR from plasmid pIJ773 (primers: SigE_HA-tag_up; SigE_HAtag_down)[63,65]. The PCR product was electroporated into E. coli BW25113/pIJ790 harboring the StE94 cosmid. E. coli cells were cultivated at 37 °C for 1 h in LB medium and plated on LB agar plate containing apramycin (50 µg/ml). Subsequently, the cosmid containing the StE94/sigE-HA gene cassette was isolated, sequenced, and transformed into the methylation defective E. coli ET12567/pUZ8002. For subsequent conjugation, the E. coli strain was mixed with heat-shocked S. coelicolor A3(2) spores, plated on an MS agar plate (Agarose, 2%; Mannitol, 2%; Soya flour, 2%; 10 mM MgCl₂) without antibiotics and incubated for 20 h at 30 °C. The plate was then overlaid with 1 ml of water containing nalidixic acid (25 µg/ml) and incubated for an additional 1 day at 30 °C. Double crossover exoconjugants (kanamycin sensitive, apramycin resistant) were selected by replica-plating of colonies onto DNA (Difco Nutrient Agar) plates containing nalidixic acid (25 µg/ml) and apramycin (50 µg/ml) with and without kanamycin (50 µg/ml)[63,66]. Insertion of the HA tag sequence downstream of the sigE was confirmed by PCR and sequencing.

**Lysozyme sensitivity test**. Strains LK3801 (wt), LK3802 (σ$^E$-HA), and LK3804 (ΔsigE) were treated with lysosyme as previously described[15]. Briefly, spores of each strain ($2.5 \times 10^7$) were platted on DNA plates and a twofold dilution series of lysozyme was immediately spotted on these lawns. Plates were photographed after two days at 30 °C.

**Cloning of promoter constructs into the pBPSA1 vector**. Upstream regions of genes SCO7657-hyaS (274 bp) and SCO4350 (188 bp), containing predicted σ$^E$-dependent promoters were PCR-amplified (primers are listed in Supplementary Table 2). Constructs where the -10 promoter regions were mutated to GGGGG were prepared synthetically (GenScript Biotech, Netherlands) or by PCR. PCR products were cloned into the pBPSA1 vector via BamHI and HindIII restriction sites upstream of the bpsA gene that encodes indigoidine (blue pigment) synthetase BpsA[31]. The resulting vectors were transformed into E. coli DH5α and sequenced with primers LK4359 and LK4360. Verified vectors were transformed into E. coli ET12567/pUZ8002 and conjugated with S. coelicolor A3(2) as described above. S. coelicolor chromosomal DNA from selected positive clones was then isolated and vector integration was verified by PCR (primers LK4360 and LK4766).

**Indigoidine extraction and measurement**. S. coelicolor strains (LK3626, LK3630, LK3633, LK3635, LK3716) containing the integrated pBPSA1 vector with promoter variants were grown as described above (Strains and growth conditions). Cells were

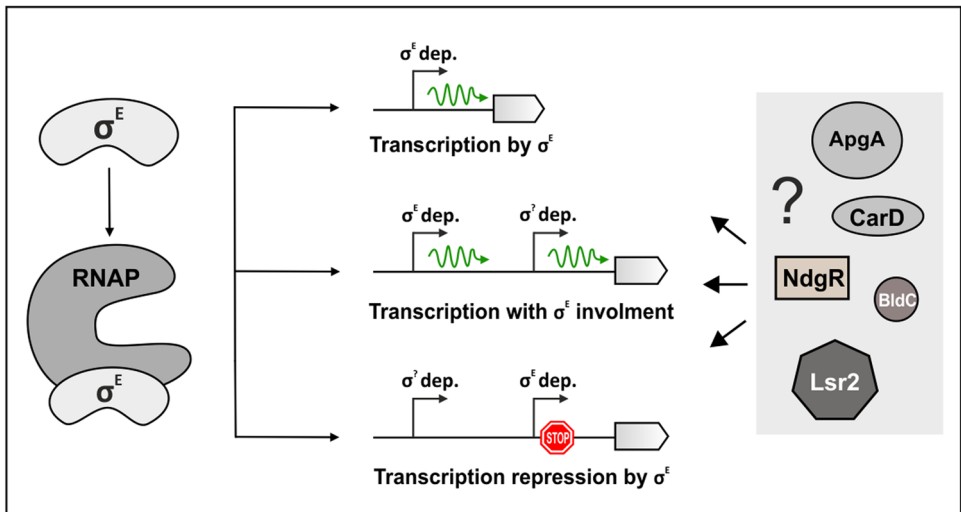

**Fig. 10 Regulation of σ^E and a model of regulation by σ^E.** σ^E binds to RNAP to form holoenzyme and binds to promoters where three possible scenarios can occur (i) σ^E is the only regulator of the gene; (ii) σ^E is one of two or more regulators; (iii) σ^E is bound to the promoter, either occluding and overlapping another promoter or forming a barrier against RNAP that has initiated transcription from upstream promoters. σ^E-dependent transcription can possibly be also modulated by additional binding partners such as ApgA, CarD, BldC, and DNA binding proteins NdgR, Lsr2.

grown for 24 h in Na-glutamate medium. 15 ml of the cell culture was then used for extraction and measurement of indigoidine[67]. Briefly, cells were centrifuged ($5000 \times g$, 10 min, RT, swinging rotor), washed once with ultrapure water, and then pelleted again ($5000 \times g$, 10 min, RT, swinging rotor). Bacterial pellets were weighed and stored at $-20\,°C$ for 2 h. Subsequently, indigoidine was dissolved by adding 1 ml of dimethylsulfoxide (DMSO) to bacterial pellets. Mixtures were vortexed for 30 s and centrifuged ($5000 \times g$, 10 min, RT, swinging rotor). Supernatants, containing indigoidine, were collected and the pigment level was measured spectrophotometrically at the 613 nm wavelength. Strain LK3716 bearing the empty integrated plasmid, was used as a negative control. The resulting data represent optical density ($OD_{613}$) per milligram of wet biomass.

**Western blot analysis.** The RNAP core cellular level was detected using mouse monoclonal antibody against RNAP beta subunit (GeneTex, cat. no: GTX12087, clone 8RB13) and secondary antibody conjugated with HRP (Sigma-Aldrich, cat. no.: A9044-2ML). σ^E-HA cellular level was detected by a high-affinity monoclonal anti-HA antibody conjugated with HRP (Roche, cat. no.: 12013819001). Bacterial growth was performed as described above with the following changes. After 23 h of growth in Na-glutamate medium, the σ^E-HA-tagged strain (LK3802) culture was equally divided into four Erlenmeyer flasks, and 20 ml from each flask was collected as non-stressed samples. Subsequently, stressors (0.2 M NaCl, 0.5 mM diamide, 4% EtOH, 42 °C) were added respectively. Stressed cultures were collected at 30 and 60 min. Bacteria were then pelleted, resuspended in 5 ml of 1×P buffer (35% NaCl; 35% $Na_2HPO_4$; 10% glycerol; v/v) containing 3 mM 2-mercaptoethanol and 1 mM serine protease inhibitor PMSF (phenylmethylsulfonyl fluoride) and sonicated $15 \times 10$-s, 0.5 amplitude, on ice (Hielscher sonicator, UP 200S). Samples were centrifuged for 15 min, at $15,000 \times g$, 4 °C. The supernatants were collected, protein concentrations measured, and samples were loaded on SDS-PAGE. Subsequently, proteins were blotted onto the PVDF membrane. The membrane was blocked with 5% non-fat milk in 1×PBS for 1 h. PVDF membrane was cut and its part containing RNAP was incubated with primary antibody at 4 °C overnight. Subsequently, the membrane was probed by HRP-conjugated secondary antibody. The second part of the

membrane, containing σ^E-HA was incubated with HRP-conjugated anti-HA antibody at 4 °C overnight. Proteins were visualized using WesternBright ECL HRP substrate (cat. no.: K-12045-D20).

**Mass spectrometry analysis of the σ^E interactome.** The experiment was performed in three biological replicates. Strains [σ^E-HA-LK3802; wt-LK3801 (negative control)] were grown as described above (Strains and growth conditions). After 23 hr of growth in Na-glutamate medium (at 30 °C), the cultures were stressed or not with 4% EtOH for 1 h. Bacteria were then pelleted, resuspended in 5 ml of 1×P buffer (35% NaCl; 35% $Na_2HPO_4$; 10% glycerol; v/v) containing 3 mM 2-mercaptoethanol and 1 mM serine protease inhibitor PMSF (phenylmethylsulfonyl fluoride) and sonicated 15×10-s, amplitude 0.5, on ice (Hielscher sonicator, UP 200 S). Samples were centrifuged for 15 min, at $15,000 \times g$, 4 °C. The supernatants were collected, and protein concentration was measured. Nuclease treatment was not used to also include proteins that bind DNA close to σ^E. Subsequently, samples were precleared for 2 h at 4 °C using Protein-G Plus Agarose (Santa Cruz sc2002) in 1×P buffer. Then, mouse anti-HA monoclonal antibody (Sigma-Aldrich, cat. no.: H3663) was added to samples and incubated overnight at 4 °C. The following day, samples were washed four times (two times with 1×P buffer, and two times with 50 mM Tris-Cl pH 7.8). Finally, samples were eluted by the addition of 50 mM Tris pH 7.8 with 4% SDS and this was followed by centrifugation ($9.000 \times g$, 5 min, RT). The samples dissolved in 4% SDS were then placed on filters (Microcon-10 kDa Centrifugal Filter) to digest overnight[68]. Digested peptides were injected onto a nanoLC (UltiMate 3000 RSLC) coupled to Orbitrap Fusion Lumos Mass spectrometer (Thermo Fisher Scientific)[69,70]. MaxQuant with Andromeda search engine (version 1.6.3.4; Max-Planck-Institute of Biochemistry, Planegg, Germany) was utilized for peptide and protein identification with databases of *S. coelicolor* (UniProtKB, Nov 27th, 2019) and common contaminants. Perseus software (version 1.6.2.3; Max-Planck-Institute of Biochemistry) was used for the label-free quantification The identified proteins were filtered for contaminants and reverse hits. Proteins detected in the data were filtered to be quantified in at least two of the triplicates in at least one condition. The data were processed to compare the

abundance of individual proteins by statistical tests in the form of a student's *t*-test and protein-abundance difference (fold-change). The final volcano plot was created using VolcaNoseR[38].

**Chromatin immunoprecipitation (ChIP).** Two strains were used: (i) strain LK3802 encoding in its native locus σ$^E$-HA, and (ii) its parental strain LK3801 without any tag (negative control). After 23 h of growth in Na-glutamate medium, the strains were stressed or not with 4% EtOH for 1 h followed by 30 min long crosslink reaction using 1% formaldehyde (CH$_2$O 36,5%–38% Sigma-Aldrich). Crosslinking was stopped with prechilled 2 M glycine for 5 ml at room temperature. Cells were centrifuged at 4 °C and washed five times with 1×PBS. The pellet was resuspended in RIPA buffer (SDS, 0.1%; sodium deoxycholate, 0.5%; Triton X-100, 0.5%; 1 mM; NaCl, 150 mM; Tris, pH 8, 50 mM; Proteases Inhibitor-Complete, Mini, ethylenediaminetetraacetic acid (EDTA)-free, Roche, 10 nM) and sonicated 6×15-s, 0.5 amplitude, on ice (Hielscher sonicator, UP 200S). Lengths of DNA fragments were analyzed by agarose gel electrophoresis (the range of DNA fragments was 200–500 bp). The cell lysate was centrifuged (20,000 × *g*, 20 min, 4 °C) and the supernatant was collected and precleared with protein-G Plus agarose (Santa Cruz sc2002) in RIPA buffer. 2 mg of total proteins were mixed with 2 µg of anti-HA high-affinity antibody (Roche, cat. no.: 11867423001) and incubated at 4 °C for 16 h. The sample was then washed three times with RIPA buffer, four times with WASH buffer (Triton X-100, 0.5%; Sodium deoxycholate, 0.4%; LiCl, 0.5 M; Tris-Cl, pH 8.5, 100 mM), two more times with RIPA buffer and twice with TE buffer (EDTA, 10 mM; Tris, pH 8, 10 mM) followed by elution with Elution buffer (SDS, 1%; EDTA,10 mM; Tris-Cl, pH 8, 50 mM). Finally, the sample was de-crosslinked by overnight incubation with 200 mM NaCl and Proteinase K at 60 °C overnight. DNA was purified by NucleoSpin gDNA Clean-up (Macherey-Nagel).

**ChIP sequencing.** ChIP-seq libraries were prepared by TaKaRa ThruPLEX DNA-Seq kit from input amounts of DNA ranging from 50 to 1000 pg. Depending on the DNA input library, amplification cycles varied from 11 to 16. Individual libraries were quantified by the Qubit fluorometer High Sensitivity DNA Assay and fragment profiles were analyzed by the Agilent 2100 Bioanalyzer High Sensitivity DNA Assay. Equimolar pools were prepared and sequenced on the Illumina NextSeq 500 platform using NextSeq 500/550 High Output Kit v2.5 (75 Cycles) including a 5% PhiX spike-in control.

**ChIP-seq—computational analysis.** ChIP-seq experiments were carried out in 4 replicates, including negative control (marked as V1–V4) and EtOH-treated (marked as E1-E4) or non-treated (marked as C1-C4) samples. Samples C1 and V4 were excluded for low sequencing quality. Sequencing data quality was checked with the FastQC (https://www.bioinformatics.babraham.ac.uk/projects/fastqc/) utility. The analysis was performed in a Chipster[71] (https://chipster.metacentrum.cz/) environment. Raw sequences were concatenated and aligned using the Bowtie2 algorithm[72] with the following parameters (--sensitive --mp 6 --np 1 --rdg 5,3 --rfg 5,3 --phred33 --no-unal -p 2). For the EtOH-treated samples, the overall alignment rate was 97.64%, for the unstressed samples the alignment rate was 97.73%. Peaks in the samples were identified using the MACS2[73] tool and an *S. coelicolor* genome sequence (available from NCBI under accession AL645882.2) against the wild type control with the following parameters: *q*-value cutoff: 0.01, Build peak model: yes, Extension size: 200, Upper M-fold cutoff: 30, Lower M-fold cutoff: 10. Identified peaks were manually checked against the wild type

control in Tablet[74] genome browser and the final set of peaks was used in the subsequent analysis.

**Promoter sequence analysis.** σ factor promoters typically contain two conserved elements, -35 and -10, separated by a spacer of 16-19 bp. We chose a strategy where we first identified these motifs individually, and then we combined the search results to form a complete motif with the given spacer. The complete motif was then searched in the region defined by the ChIP-seq peak borders. To derive the complete binding motif, we extracted a 100 bp sequence region centered on ChIP-seq peak summits from peaks. Then we used MEME[75] (meme -revcomp -dna -mod zoops -minw 4 -maxw 7 -allw -nmotifs 10) to find the best scoring motif and identified it as the -10 binding motif part. To find the -35 element, we extracted 10 bp segments starting 23 bp upstream from the discovered -10 motif sites. Then we analyzed the extracted sequences with MEME (meme -mod zoops -nmotifs 3 -minw 4 -maxw 8 -dna -allw) and the best scoring motif was selected as the -35 element. Consequently, we built a composite motif, containing both the -35 and -10 motif parts, from the sequences containing both. The composite motif was created for four different lengths of the spacer region (16-19 bp). The composite motif was then used to search the σ$^E$ ChIP-seq peak sequences as defined by the ChIP-seq peak borders (see Supplementary Data 1 for the location of individual peaks). The search was done using FIMO[76] with a *p*-value threshold of 0.05, the found motif sites were then manually checked for consistency.

To analyze occurrences of the HrdB binding sites in the σ$^E$ ChIP-seq peak sequences, we used FIMO (*p*-value threshold of 0.05) with a previously identified motif for HrdB[4] from position 3 to position 12 (conserved motif core). The found motif sites were further filtered using the following criteria: 1) motif with the lowest *p*-value on each sequence were retained and 2) all motif sites with the sequence matching "--TA---T--" were retained ('-' denotes any base).

The identification of BldN and σ$^R$ binding sites in the SCO4350 upstream sequence was done based on known consensus sequences[19,56]. For BldN, we obtained three similar motifs (two with varying spacer) and two motifs with varying spacer for σ$^R$[77]. These motifs were identified with FIMO in the SCO4350 upstream region (4766910-4767085) with *p*-values of 0.001 and less. The analyzed sequence is shown in Supplementary Fig. 4C. Based on experimental validation (qPCR and indigoidine quantification), we selected candidates that were positively (*hyaS*, *ssgD*, SCO3396) or negatively (SCO4350, *accB*, SCO4921, *accA2*) regulated by σ$^E$. We created motifs of -10 and -35 regions for positively and negatively regulated genes.

**Kinetic modeling (data acquisition).** We used gene expression time series published by Nieselt et al.[14], downloaded from GEO with accession number GSE18489. The time series contains gene expression measurements from 32-time points without replicates (20, 21, 22, 23, 24, 26, 27, 28, 29, 30, 31, 32, 33, 34, 35, 36, 37, 38, 39, 40, 41, 42, 43, 44, 46, 48, 50, 52, 54, 56, 58, 60 h). The data that had been published in the log$_2$ scale were, for the following analysis, exponentiated to give original microarray reads. As the Nieselts experiments were run without replicates, the variance of the expression profiles is very high. In order to find the trend in expression, the time series were smoothed with Sawitzky-Goaly filter (Matlab function smoothdata, method sgolay). For further computations, the smoothed profiles were subsampled to a 24 min interval. The modeling step using the time series is described in the following paragraph. The raw and interpolated data were saved in Supplementary Data 2.

**Gene expression modeling**. In order to find whether the ChIP-seq-identified genes are also controlled by σ[E] under the conditions defined in the time series of gene expression, which represent regular growth, we used an ODE-based gene expression model. The model was originally defined in[78] and recently used in[53], where also the details of the algorithm can be found. Briefly, we assume that there exists a nonlinear relationship between the σ factor and the regulated gene expression levels expressed as an ordinary differential equation including the target mRNA degradation. The model uses gene expression time series, and the optimization procedure alters iteratively the initial parameter estimates to fit the modeled regulated gene expression profile to the measured one. If the procedure, under certain constraints that maintain the biological plausibility of the model, is able to fit the measured target gene expression profile, the regulation of the given gene by the given σ factor is considered possible. If the fit cannot be reached, the regulation is disproved. A crucial parameter is the weight $w$, which weighs the influence of the given regulator on the expression of the regulated gene. Its sign indicates whether the regulatory effect is positive (stimulation) or negative (repression). The computation was performed using a simple Euler integration with a simulated annealing optimizer. The modeling was repeated 256 times with random values of initial parameter estimates for each target gene expression profile and the respective σ factor. From the 256 runs, the model giving the smallest sum of squared differences between the measured and computed expression profiles was selected. We used a procedure published in[79] using the command line version running on a graphic card and executed from a Matlab script. The script executables and a corresponding Cytoscape plugin can be downloaded from https://github.com/cas-bioinf/genexpi/wiki. Parameters of the models computed for the genes in Supplementary Tables 4 and 5 are in Supplementary Data 3.

**RNA isolation and RT-qPCR**. The wt (LK3801) and Δσ[E] (LK3804) strains were used for qPCR analyses of 13 selected genes under normal conditions (w/o stress) or EtOH treatment. Bacteria were shaken for 23 h (corresponding to exponential phase[64] at 30 °C, 250 rpm, and pH 7 was maintained during the growth. After 23 h of growth in Na-glutamate medium, the strains were stressed or not with 4% EtOH for 1 h. RNA was isolated from 2 ml of bacterial culture (at $OD_{600} = 0.8$) using RNeasy Mini Kit (QIAGEN, Cat. No.: 74004) according to manufacturer's protocol with two minor changes: (i) During the addition of RLT buffer (included in the kit), 10 ng of spike RNA were added to each sample. The spike RNA was a fragment of Plat mRNA (718 bp) from *M. musculus* prepared from pJET_-Plat_IVTs plasmid (kind gift from Dr. Radek Malík) using T7 Ribomax Express Large Scale (Promega)]; (ii) after the addition of RLT buffer and spike RNA, the sample was sonicated twice (0.5 amplitude, Hielscher sonicator, UP 200 S) for 30 s with a 1-min pause on ice between sonications. Isolated RNA was treated by DNA-free™ DNA Removal Kit (Invitrogen, Cat. No.: AM1906). RNA was reverse-transcribed using random hexamers and SuperScript™ III Reverse Transcriptase (Invitrogen, Cat. No.: 18080044). Subsequently, qPCR was done using the Light Cycler LC480 (Roche Diagnostic, Mannheim, Germany). Primers for qPCR are listed in Table 2. 2 μl of each cDNA sample (including RNA samples without reverse transcription as controls to confirm successful DNase treatment), in duplicates were spotted in the plate wells together with 2.5 μl of SYBR Green Master mix (Roche Diagnostic, Mannheim, Germany) and 0.5 μl of respective primers (total volume of 5 μl). PCR Temperature conditions: 1 × 95 °C for 7 min followed by 45 x [95 °C for 20 s, 61 °C for 20 s and 72 °C (in the single acquisition mode) for 35 sec]; this was

followed by 1 melting curve cycle at 95 °C for 15 s, 55 °C for 1 min 1 s, and 95 °C (in the continuous acquisition mode with 10 acquisitions/°C) and 37 °C for 1 min 1 s. Primer efficiency was tested using the standard curve with known, increasing amounts of *S. coelicolor* chromosomal DNA. The $\Delta C_t$ method[80] was used to determine the relative quantities of cDNAs. The absolute quantities were then calculated from sample $C_t$ values compared to the values obtained from the standard curve. The values were then normalized to spike RNA values and cell optical density.

**Statistics and reproducibility**. All data were collected and analyzed using standard statistical methods. Information about the statistical analyses used is included, when relevant, in figure legends and in Supplementary Data 5.

**Reporting summary**. Further information on research design is available in the Nature Portfolio Reporting Summary linked to this article.

## Data availability

ChIP-seq data and technical details were deposited at ArrayExpress (https://www.ebi.ac.uk/biostudies/arrayexpress) under the accession number E-MTAB-10987. Visualization of ChIP-seq results is available at https://cas-bioinf.github.io/Scoe_SigE_ChIP-seq/ (https://doi.org/10.5281/zenodo.10351341). Code for promoter binding motif analysis is available at https://github.com/cas-bioinf/Scoe_SigE_ChIP-seq (https://doi.org/10.5281/zenodo.10351204). The mass spectrometry proteomics data were deposited to the ProteomeXchange Consortium via the PRIDE [1] partner repository with the dataset identifier PXD041877. Source data that are used in charts are available in Supplementary Data 5. Supplementary Tables 2–6 can be found in Supplementary Data 6.

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

## Acknowledgements

We thank the Core Facility of Genomics and Bioinformatics at IMG for its excellent support. We thank Prof. M. Buttner for kindly providing $\sigma^E$ deletion strain of *S. coelicolor*, Dr. R. Malík for the pJET_Plat_IVTs vector, and Dr. J. Kormanec for the pBPSA1 vector. The work was supported institutionally by RVO:61388971, ELIXIR CZ research infrastructure project (MEYS Grant No: LM2023055) including access to computing and storage facilities to J.V., M.S., M.K., Michal H., and A.Z., Czech Science Foundation (No. 22-06342K; to L.K.) and the project National Institute of Virology and Bacteriology (Programme EXCELES, ID Project No. LX22NPO5103)—Funded by the European Union—Next Generation EU to L.K., J.P. and D.V. CAS Programme to support prospective human resources—post Ph.D. candidates (L200202151) to J.P. AK and Martin H. were institutionally supported by RVO: 68378050.

## Author contributions

J.P., L.K., and J.V. conceptualized the project. J.P., A.Z., and D.V. performed the experiments. M.S and J.V. performed gene expression modeling and computational analyses. Michal H. and M.K. prepared ChIP-seq libraries and performed sequencing. Martin H. and A.K. performed mass spectrometry analysis. J.V., J.P., and L.K. wrote the first draft of the manuscript. All authors read and contributed to the final version.

## Competing interests

The authors declare no competing interests.
