## [Peer Review File · Communications Biology]

Reviewers' comments:

Reviewer #1 (Remarks to the Author):

To authors.

The text is didactically written, easy and pleasant to read. The data are clear, elegantly presented, and in line with the conclusions. Authors demonstrate in this work that σE (SigE), a sigma factor, can repress transcription at specific promoters. This result is unexpected as the general schematic microbiologist have in mind, in gram positive or negative bacteria, is that SigE is induced under stressful conditions operating affecting the integrity of the bacterial wall, and SigE activates specific promoters. In *S. coecolor*, this model is partial, and at certain promoters SigE forms a complex with RNAP and additional factors 'blocking' the transcription initiation complex (Fig. 9). Data are of interest for the microbiologist community, especially for people involved in gene expression control and bacterial adaptation.

I do not have any key comment to significantly improve the manuscript; only details.

-I am not familiar with *S. coecolor* and I cannot appreciate how difficult this would be to do experimentally: although -35 boxes are different, what about swapping -10 boxes in constructs used Fig.7? The use of chimeric promoters could reinforce the fact that SigE's positive or negative action on transcription initiation is primarily a function of SigE's interaction with key nucleotides recognized on the DNA, the -10 box. Replacing the -10 boxes of SCO7657 with that of SCO4350 (actually only 4 nucleotides), and vice versa, could reverse the patterns obtained in Fig. 7A and 7B. This assumes that the -10 box sequence is an essential discriminant of SigE's duality of action. Indeed, the open question is what causes a promoter to be activated or repressed by SigE? I suppose the authors have already thought of it.

-It would be more practical for microbiologists if the time axis were indicated in hours (e.g., Fig 2, 3, 5, 6).

Francis Repoila

Reviewer #2 (Remarks to the Author):

The manuscript by Pospisil et al describes a putative repressing function of SigE in *Streptomyces coelicolor*, in addition they present the associated regulon and a potential anti-sigma factor ApgA. Together with the repressing function, the suggested anti-sigma function of ApgA and interaction with other transcription factors is new information, although the latter two are of more preliminary character. The SigE regulon has been reported previously, although the regulon presented in this study to some extent differ, due to alternative stimuli and conditions. In general, the study is sound when it comes to experimental approaches, where for example the inhibitory function is demonstrated with RT-qPCR for a couple of genes and with promoter mutations for one of the genes. However, although the discussion part is interesting and shows the complexity for gene regulation involving multiple transcription and sigma factors, the way the study is presented leaves many questions with regard to the title, which can make it difficult for readers to appreciate the findings, there are things that could be better explained related to biological contexts and functional roles.

Comments are specified below.

Include information of how this EtOH concentration affect *S. coelicolor* growth, membrane permeability etc in supplementary data. Also for better understanding and biological interpretation, provide information of which temperature is used for EtOH treatment, indicate temp and EtOH conc (%) in

figure legend for figure 1.

Include description of genes (not just locus tags) when presenting the genes that were repressed (page 8) to increase biological understanding/promote new ideas to arise. Some of the locus tags are explained in the discussion section, but if they are mentioned already in the result part, interpretations are facilitated. Similar for the genes selected for RT-qPCR analyses (page 10-11), Give information about the gene products and also explain why these genes were selected for this analysis, was it based on peak characteristics, nature of the gene product that made them interesting to test, or randomly picked?

In this study, only ChIP peaks situated in potential promoter regions upstream coding sequences are considered. Some other ChIP-seq studies of other alternative sigma factors also report fraction of intragenic binding of sigma factors as well as binding to the anti-sense strand have been reported (Bonocora RP, et al PLoS Genet 11:e1005552; Samuels DJ, et al 2013. BMC Genomics 14:602; Mahmud AKMF, et al 2020. mSystems 5:e01006-20). For general knowledge of sigma factors, It would be important to know if this is the case also for SigE? Potentially, intragenic binding should in some cases cause block of transcription, and there have been reports of inhibitory effects of sigma factors binding to the anti-sense strand (Mahmud et al. mSystems2020). This needs to be discussed.

Kinetic modeling based on data from fermentor grown *S. coelicolor* showing metabolic switches is frequently used by this research group for correlations and clustering of gene expression groups. This is fine, but without explanation it is confusing since the genes in focus in this study are induced by EtOH after much shorter time. So please provide informative explanation and also some interpretations, discuss with regard to the distinct expression patterns seen for sigE versus hrdb regulated genes with relation to their co-expression with other genes in the fermentor study. Although relatively minor changes, what does it say about SigE to be almost 2-fold more expressed after 30h, what is the phenotype of the culture in this phase, can it be coupled to surface changes etc.

It is suggested that HrdB and SigE work together, but also here the biological discussion is very limited. Can it be that HrdB have stronger binding, and that SigE come in under certain stresses? Is there any correlation to expression of its anti-sigma factors. Sigma factor competition has been discussed by others (Mauri M, Klumpp S. 2014. A model for sigma factor competition in bacterial cells. PLoS Comput Biol10:e1003845)

minor

-coelicolor in main title shall be in italic

Reviewer #3 (Remarks to the Author):

The team aimed at revealing the *Streptomyces coelicolor* σ^E regulon genes after EtOH stress treatment. Using ChIP-seq, RT-qPCR, and pull-down MS, authors find σ^E both as a direct activator or repressor of transcription, and some anti-sigma factors may be involved in these processes.

The article discovered some functional mechanisms of σ^E in *Streptomyces coelicolor*, but did not establish a relationship between activator or repressor of regulated genes and stress responses. The data on protein interaction lacks detailed analysis and verification, and this data is not sufficiently related to the topic of the article.

Major

1. Line 100 Fig1. σE was rapidly degraded during osmotic, oxidative, and heat stresses. From ChIP-seq and qPCR data, σE inhibits some genes. So, when the σE protein decreases, some gene transcription increases. How do these genes increase bacterial stress tolerance?
2. Line 117. ChIP-seq was performed in the absence/presence of EtOH. Why not display the overall peak differences and specific site peak differences (using tools such as IGV, ChIPseeker...), so that we can visually see the binding regulation differences of σE under EtOH pressure.
3. Line 170, What is the importance of the three genes and what is their roles in being inhibited by σE ? In Fig 4-5-6, The author provides detailed evidence that σE binding DNA may activate and inhibit its transcription. What is its physiological significance?
4. Line 306. Why not mutate the -35 region? The -35 region of the two genes is different.
5. Line 324 Fig 8. Why not generate a more conservative logo with a large number of DNA?
6. Line 442 Fig10. The diagram does not highlight the core findings of the paper. The paper's name is " σE of *Streptomyces coelicolor* can function both as an activator and repressor of transcription", focusing on the "activator" and "repressor". However, this figure depicts TCS CseC/B regular sigE, BldG is an anti-anti-sigma factor. Is there any actual empirical evidence in the whole paper to prove this? Are they related to the stimulation of EtOH? Does EtOH cause phosphorylation changes of CseB, then regulate σE ?

Minor

7. Line 109. "ChIP-Seq data analysis" is the method name, not a suitable result name. Many subheadings have similar issues.
8. Line 470. 0,1 or 0.1
9. Line 726. Chip or ChIP
10. Line 610. Why after "ChIP sequencing" is "RNA isolation and, RT-qPCR. Line 638. The "ChIP-seq" method should conclude ChIP, sequence, and data analysis.
11. Line 750. All references should be carefully revised. such as species and gene names should be italicized, the display of authors' names is inconsistent, capitalization issues, and incomplete page number information.

Reviewer #4 (Remarks to the Author):

This study analyses the genomic binding region of Sigma E in *Streptomyces coelicolor* and identifies a new regulatory target. The authors considered regulation by Sigma E based on transcriptome analysis. In particular, the authors have found genes that are repressed and proposed regulatory mechanisms. As the Sigma factor is essentially a transcription initiator, it would be unusual and academically novel if it acts as a repressor. However, the disparate conditions of the different experiments in this study do not allow for comparison or discussion. In addition, the only basis for the claimed molecular mechanism is the gene expression pattern, and the molecular mechanism itself has not been demonstrated at all. The authors' claims are premature because the conditions of observation across different experiments are not unified and the molecular mechanism is not demonstrated.

Major comments

1. Why was the sigma E-deficient strain not used as a negative control in ChIP-Seq analysis? Although the comparison is made with and without the addition of EtOH, Sigma E was also expressed under non-additive conditions, as shown in Fig. 1. Therefore, the demonstration of the specificity of ChIP for Sigma E is insufficient and must be performed with the sigma E-deficient strain; the samples for the input of ChIP analysis conditions should also be clearly stated in the Methods section.
2. The ChIP-Seq data analysing Sigma E and the gene expression pattern in Figure 3 must be

validated under identical conditions. As shown in Figure 1, the effect of stress on the expression level of Sigma E was different, and the data were not from the same stress and the same timing, which makes it meaningless to consider it as an effect of Sigma E. In addition, the time scales in the sampling were completely different and therefore not comparable.

3. Transcriptional regulation builds a network and hierarchical structure, with sigma factors and transcription factors regulating each other. Therefore, in order to discuss molecular mechanisms, the directness of gene expression regulation must be demonstrated, which is fatal to the lack of this study throughout.

4. The proposed molecular mechanisms are based only on the mRNA level, which is insufficiently validated and only speculative. For example, the models for Lines 221-229, Lines 230-242 and Lines 274-280 must be demonstrated to be direct effects and the possibility of indirect effects must be excluded.

5. Line 292-: states that mutations in the promoter sequence have resulted in Sigma E no longer binding, but this is speculative. It must be demonstrated that binding is actually altered. For example, it needs to be verified whether Sigma E and Sigma R are being replaced, as the authors would like to claim.

6. The authors claim that the Sigma E model requires Sigma E binding by itself or Sigma E holo enzyme to be stopped by the promoter, but has this been demonstrated? The Sigma N holo enzyme of *E. coli* can act as a repressor as it stops to the formation of a closed complex in the absence of an enhancer, but what about Sigma E in *S. coelicolor*? If there are findings, they should be cited.

Minor comments

7. Is the decrease in protein levels in Sigma E under NaCl, diamide and 42°C stresses in Fig. 1 a valid phenomenon? Relevant previous findings should be cited and explained.

8. Why are SCO2970 and SCO6056 listed as Periplasmic/exported/lipoproteins in Functional Category in Table S3 when their Name is hypophetical protein? In other genes, hyphetical protein is an Unknown function.

9. It is difficult to understand the credibility of the target list; the binding strength (ratio?) of the ChIP should be shown in Tables S3-S6.

10. Similarly, consensus sequences and conservation (%) should also be indicated in Table S3-S6.

11. Known and novel targets should be indicated in Table S3-S6.

12. Line 127: It would be easier to indicate previously identified sequences if they are also described and shown in the text.

13. The blank background in Table S4-S6 should be described as an indirect effect, as no Sigma E binding was observed.

14. The phase shift in Fig 3 does not distinguish between indirect and direct effects.

15. Line 153: With regard to the six sRNAs, how were they modelled (grouped) in the absence of kinetic data?

16. The timing and conditions of the expression analysis in Figure 4 are unclear.

17. The effect of the presence or absence of EtOH addition should also be examined in Figure 7.

18. The consensus sequence in Figure 8 is not objective and cannot be discussed because the number of subjects is too small.

19. For objective evaluation in interaction experiments with Sigma E, gel images of the SDS-PAGE samples after pull-down should be shown.

20. As mentioned above, it has been demonstrated that Sigma N in *E. coli* acts as a repressor (DOI: 10.1099/mgen.0.000653). This reference must be cited in this paper as it is an example of RNAP holo working in repression.

Reviewers' comments:

Reviewer #1 (Remarks to the Author):

To authors.

The text is didactically written, easy and pleasant to read. The data are clear, elegantly presented, and in line with the conclusions. Authors demonstrate in this work that σ^E (SigE), a sigma factor, can repress transcription at specific promoters. This result is unexpected as the general schematic microbiologist have in mind, in gram positive or negative bacteria, is that SigE is induced under stressful conditions operating affecting the integrity of the bacterial wall, and SigE activates specific promoters. In *S. coecolor*, this model is partial, and at certain promoters SigE forms a complex with RNAP and additional factors 'blocking' the transcription initiation complex (Fig. 9). Data are of interest for the microbiologist community, especially for people involved in gene expression control and bacterial adaptation.

I do not have any key comment to significantly improve the manuscript; only details.

-I am not familiar with *S. coecolor* and I cannot appreciate how difficult this would be to do experimentally: although -35 boxes are different, what about swapping -10 boxes in constructs used Fig.7? The use of chimeric promoters could reinforce the fact that SigE's positive or negative action on transcription initiation is primarily a function of SigE's interaction with key nucleotides recognized on the DNA, the -10 box. Replacing the -10 boxes of SCO7657 with that of SCO4350 (actually only 4 nucleotides), and vice versa, could reverse the patterns obtained in Fig. 7A and 7B. This assumes that the -10 box sequence is an essential discriminant of SigE's duality of action. Indeed, the open question is what causes a promoter to be activated or repressed by SigE? I suppose the authors have already thought of it.

Response:

It is a valid suggestion. It likely depends on the interaction between RNAP and the promoter. We can speculate that in the case of the inhibitory function of σ^E the interaction enables formation of a stable RNAP-DNA complex that is compromised in its ability to initiate transcription/leave the promoter. Mechanistic details of this phenomenon will be addressed in a future project.

-It would be more practical for microbiologists if the time axis were indicated in hours (e.g., Fig 2, 3, 5, 6).

Response:

We agree.

Action taken:

We changed the labeling of x-axes to hours in **Figures 1, 3-6** and in **Supplementary Figures S3-4**.

Francis Repoila

Reviewer #2 (Remarks to the Author):

The manuscript by Pospisil et al describes a putative repressing function of SigE in *Streptomyces coelicolor*, in addition they present the associated regulon and a potential anti-sigma factor AppA. Together with the repressing function, the suggested anti-sigma function of AppA and interaction with other transcription factors is new information, although the latter two are of more preliminary character. The SigE regulon has been reported previously, although the regulon presented in this study to some extent differ, due to alternative stimuli and conditions. In general, the study is sound when it comes to experimental approaches, where for example the inhibitory function is demonstrated with RT-qPCR for a couple of genes and with promoter mutations for one of the genes. However, although the discussion part is interesting and shows the complexity for gene regulation involving multiple transcription and sigma factors, the way the study is presented leaves many questions with regard to the title, which can make it difficult for readers to appreciate the findings, there are things that could be better explained related to biological contexts and functional roles.

Comments are specified below.

Include information of how this EtOH concentration affect *S. coelicolor* growth, membrane permeability etc in supplementary data.

Response:

We did not measure growth/membrane permeability during the experiment. Nevertheless, in *Streptomyces albus*, it was reported that after EtOH addition mycelia became thicker and more branched. Moreover, this treatment increased the intracellular ROS level and the ratio of un/saturated fatty acids in the cell membrane. This could improve fluidity of the cytoplasmic membrane (**PMID: 34670188**). In *Streptomyces venezuelae*, changes in the transcriptome were described after EtOH treatment, revealing a number of σ -factor encoding genes whose expression was changed after the treatment, including a σ^E family σ factors such as *vnz_RS00060*, *vnz_RS15950*, *vnz_RS18620*, *vnz_RS22785*, *vnz_RS23885* (**PMID: 36314940**)

Action taken:

We included the information about EtOH treatment of *S. venezuelae* and *S. albus* (**page 22, line 391**).

Also for better understanding and biological interpretation, provide information of which temperature is used for EtOH treatment, indicate temp and EtOH conc (%) in figure legend for figure 1.

Response:

We agree.

Action taken:

EtOH concentration (as well as those of diamide, NaCl) and the temperature are now specified in the legend of **Figure 1**.

Include description of genes (not just locus tags) when presenting the genes that were repressed (page 8) to increase biological understanding/promote new ideas to arise. Some of the locus tags are explained in the discussion section, but if they are mentioned already in the result part, interpretations are facilitated. Similar for the genes selected for RT-qPCR analyses (page 10-11), Give information about the gene products and also explain why these genes were selected for this analysis, was it based on peak characteristics, nature of the gene product that made them interesting to test, or randomly picked?

Response:

We agree, information about gene product functions will help.

Action taken:

The information about gene product functions is now provided on **pages 8 and 10**. The selection of the genes for RT-qPCR was not random and it is described in the text (**pages 10-12 and 16**).

In this study, only ChIP peaks situated in potential promoter regions upstream coding sequences are considered. Some other ChIP-seq studies of other alternative sigma factors also report fraction of intragenic binding of sigma factors as well as binding to the anti-sense strand have been reported (Bonocora RP, et al PLoS Genet 11:e1005552; Samuels DJ, et al 2013. BMC Genomics 14:602; Mahmud AKMF, et al 2020. mSystems 5:e01006-20). For general knowledge of sigma factors, it would be important to know if this is the case also for SigE? Potentially, intragenic binding should in some cases cause block of transcription, and there have been reports of inhibitory effects of sigma factors binding to the anti-sense strand (Mahmud et al. mSystems2020). This needs to be discussed.

Response:

This is a very valid comment. Intragenic binding of σ factors was already shown previously where RNAP can associate with various σ factors, depending on promoter-like sequences within genes. This has regulatory functions (**PMID: 35653571**). In our ChIP-seq experiment, almost half of the σ^E binding sites (92 out of 193) were intragenic.

Action taken:

We added the information about intragenic peaks to **Supplementary File 1** (first sheet: Column named as "*summit_in_gene_annot*"). The predicted σ^E binding motifs with the lowest *p*-value are also included in **Supplementary File 1** (second sheet "*Intragenic peaks only*"). According to this, we modified text of ChIP-seq results (**page 5, line 109**) and Discussion (**page 23, line 408**) and cite relevant papers.

Kinetic modeling based on data from fermentor grown *S. coelicolor* showing metabolic switches is frequently used by this research group for correlations and clustering of gene expression groups. This is fine, but without explanation it is confusing since the genes in focus in this study are induced by EtOH after much shorter time. So please provide informative explanation and also some interpretations, discuss with regard to the distinct expression patterns seen for sigE versus hrdb regulated genes with relation to their co-expression with other genes in the fermentor study.

Response:

The fermentor study was used to find correlations between regulators and their targets. This was possible because the fermentor study was unprecedentedly detailed (32 time points) thereby allowing this type of analysis. The results (identified regulators-target genes) could then be applied also to other conditions. This was subsequently tested for selected genes (RT-qPCR), confirming validity of the approach.

Action taken:

We now mention the rationale behind using the expression dataset for kinetic modelling in the text **(page 7, line 146)**.

Although relatively minor changes, what does it say about SigE to be almost 2-fold more expressed after 30h, what is the phenotype of the culture in this phase, can it be coupled to surface changes etc. It is suggested that HrdB and SigE work together, but also here the biological discussion is very limited. Can it be that HrdB have stronger binding, and that SigE come in under certain stresses?

Response:

It suggests that in the fermentor, over time, some stress was generated, and this correlated with an increase in σ^E expression concomitant with a decrease in HrdB expression. The reciprocal relationship between σ^E and HrdB levels (at least within the first 30 hrs) likely affects the competition of the two factors for the RNAP core.

Action taken:

This aspect is now discussed in Discussion **(page 24, 445)**.

Is there any correlation to expression of its anti-sigma factors. Sigma factor competition has been discussed by others (Mauri M, Klumpp S. 2014. A model for sigma factor competition in bacterial cells. PLoS Comput Biol10:e1003845)

Response:

Generally, for the anti- σ factors expressed in the same operon as the σ factor, their correlation with the σ factor is high. An example of modeling of the influence of the regulatory network of the σ factor including its anti- σ factors was shown in our recent paper **(PMID: 36552239)**.

Out of the σ factors studied in our work, HrdB has no known anti- σ factor. In this study, we identified ApgA as a potential anti- σ factor, associating with σ^E in non-stressed conditions. The *apgA* gene was also identified previously as a member of σ^E regulon **(PMID: 30907454)**. However, the kinetic modeling of *apgA* expression does not correlate with the expression profile of σ^E . It suggests that the expression of *apgA* could be driven by more than one σ factor.

minor

-coelicolor in main title shall be in italic

Action taken:

Done

Reviewer #3 (Remarks to the Author):

The team aimed at revealing the *Streptomyces coelicolor* σ^E regulon genes after EtOH stress treatment. Using ChIP-seq, RT-qPCR, and pull-down MS, authors find σ^E both as a direct activator or repressor of transcription, and some anti-sigma factors may be involved in these processes.

The article discovered some functional mechanisms of σ^E in *Streptomyces coelicolor*, but did not establish a relationship between activator or repressor of regulated genes and stress responses. The data on protein interaction lacks detailed analysis and verification, and this data is not sufficiently related to the topic of the article.

Major

1. Line 100 Fig1. σ^E was rapidly degraded during osmotic, oxidative, and heat stresses. From ChIP-seq and qPCR data, σ^E inhibits some genes. So, when the σ^E protein decreases, some gene transcription increases. How do these genes increase bacterial stress tolerance?

Response:

SCO5981, SCO5535-*aacB*, SCO4921-*accA2*, SCO4350 were shown (using RT-qPCR and/or promoter mutation analysis) as genes that are negatively regulated by σ^E . SCO5981 is a hypothetical protein with unknown function. The genes *aacB* and *accA2* encode enzymes that are involved in conversion of acetyl-CoA to malonyl-CoA which are precursors for synthesis of antibiotic actinorhodin and bioactive compound undecylprodigiosin (**PMID: 16950896**, **PMID: 24562326**). Moreover, it was shown that undecylprodigiosin stimulates programmed cell death in *S. coelicolor* (**PMID: 30127771**). We did not test if these genes are transcribed during osmotic, oxidative, and heat stresses. We have information about their expression in non-stressed conditions or EtOH treatment.

Previously, it was shown that an actinorhodin deficient strain is far more sensitive to diamide stress, suggesting that actinorhodin could have anti-oxidative properties (**PMID: 32444655**). In our work, diamide stress leads to rapid degradation of σ^E that could result in derepression of actinorhodin pathway genes *aacB* and *accA2*.

Action taken:

A correlation between gene function, degradation of σ^E after *e. g.* diamide stress, and the negative regulatory role of σ^E is now mentioned in Discussion (**page 25, line 457**).

2. Line 117. ChIP-seq was performed in the absence/presence of EtOH. Why not display the overall peak differences and specific site peak differences (using tools such as IGV, ChIPseeker...), so that we can visually see the binding regulation differences of σ^E under EtOH pressure.

Response:

Good point, thank you.

Action taken:

We added a **Supplementary Figure S2**, showing the whole genome view of normalized read coverage for respective sample groups. To assess the differences in detail, we have prepared an **interactive genome browser** (<http://78.128.246.104:8081/>) **user: rev** and **password: Ahy8q5Y4c0J4E**.

3. Line 170, What is the importance of the three genes and what is their roles in being inhibited by σ^E ?

Response:

These genes were modeled by kinetic modeling and one of them (SCO4350) was experimentally confirmed as repressed by σ^E . By RT-qPCR, we found other three genes (SCO5981, SCO5535-*aacB* and SCO4921-*accA2*) that are also negatively regulated by σ^E . See also response to your comment #1.

SCO4350 is a putative integrase, containing the SAM domain that is able to cleave DNA. Hypothetically, repression of SCO4350 prevents its expression during non-stressed conditions that could result in changes of chromosome composition (**PMID: 9082984**).

As mentioned above. The genes *aacB* and *accA2* encode enzymes that are involved in metabolism of actinorhodin and undecylprodigiosin (**PMID: 16950896, PMID: 24562326**).

Undecylprodigiosin stimulates programmed cell death in *S. coelicolor* (**PMID: 30127771**).

Therefore, σ^E -mediated repression of undecylprodigiosin production (in non-stressed conditions) could protect the cell from accumulation of this metabolite, which can lead to cell death.

Action taken:

We added information about potential significance of σ^E -mediated gene repression into Discussion and cited relevant papers (**page 25, line 450**).

In Fig 4-5-6, The author provides detailed evidence that σ^E binding DNA may activate and inhibit its transcription. What is its physiological significance?

Response:

Yes, more detailed insight into the function and significance of selected genes will be useful.

Action taken:

We added more detailed descriptions of physiological and biological significance of σ^E controlled genes into the chapters ***Identification of positively and negatively σ^E -regulated genes*** (**page 10**) and Discussion.

4. Line 306. Why not mutate the -35 region? The -35 region of the two genes is different.

Response:

It is possible. However, the simplest approach to prevent σ^E binding is to mutate the -10 region as we did. Published (**PMID: 24268774**) as well as our unpublished data with mutations of -35 regions (different promoters in different organisms) showed that such promoters could still retain some activity, complicating interpretation of the experiment.

5. Line 324 Fig 8. Why not generate a more conservative logo with a large number of DNA?

Response:

A large number of sequences (using promoter sequences that were not validated experimentally) can potentially introduce some noise that would diminish the information value of the logo. Therefore, we prefer to keep the current format of the logo.

Action taken:

We now explicitly state the number of sequences to create the logo in the text (**page 20, line 346**)

6. Line 442 Fig10. The diagram does not highlight the core findings of the paper. The paper's name is " σ E of *Streptomyces coelicolor* can function both as an activator and repressor of transcription", focusing on the "activator" and "repressor". However, this figure depicts TCS CseC/B regular sigE, BldG is an anti-anti-sigma factor. Is there any actual empirical evidence in the whole paper to prove this? Are they related to the stimulation of EtOH? Does EtOH cause phosphorylation changes of CseB, then regulate σ E?

Response:

We agree.

Action taken:

We modified the **Figure 10**, (i) removing speculative claims about the potential anti- σ factor, and (ii) removing the upper part of the figure that showed the upstream regulatory events already described in the literature.

Minor

7. Line 109. "ChIP-Seq data analysis" is the method name, not a suitable result name. Many subheadings have similar issues.

Response:

We agree.

Action taken:

We changed most of the subheadings as suggested.

8. Line 470. 0,1 or 0.1

Response:

The latter.

Action taken:

We changed "0,1" to "0.1"

9. Line 726. Chip or ChIP

Response:

The latter.

Action taken:

We changed Chip to ChIP.

10. Line 610. Why after "ChIP sequencing" is "RNA isolation and, RT-qPCR. Line 638. The "ChIP-seq" method should conclude ChIP, sequence, and data analysis.

Response:

Thank you for pointing this out, it was a mistake.

Action taken:

We changed the order of methods. RNA isolation and RT-qPCR are now at the end of this section.

11. Line 750. All references should be carefully revised. such as species and gene names should be italicized, the display of authors' names is inconsistent, capitalization issues, and incomplete page number information.

Response:

We agree. However, the format was created by End Note.

Action taken:

We manually corrected the formatting mistakes.

Reviewer #4 (Remarks to the Author):

This study analyses the genomic binding region of Sigma E in *Streptomyces coelicolor* and identifies a new regulatory target. The authors considered regulation by Sigma E based on transcriptome analysis. In particular, the authors have found genes that are repressed and proposed regulatory mechanisms.

As the Sigma factor is essentially a transcription initiator, it would be unusual and academically novel if it acts as a repressor. However, the disparate conditions of the different experiments in this study do not allow for comparison or discussion. In addition, the only basis for the claimed molecular mechanism is the gene expression pattern, and the molecular mechanism itself has not been demonstrated at all. The authors' claims are premature because the conditions of observation across different experiments are not unified and the molecular mechanism is not demonstrated.

Major comments

1. Why was the sigma E-deficient strain not used as a negative control in ChIP-Seq analysis? Although the comparison is made with and without the addition of EtOH, Sigma E was also expressed under non-additive conditions, as shown in Fig. 1. Therefore, the demonstration of the specificity of ChIP for Sigma E is insufficient and must be performed with the sigma E-deficient strain; the samples for the input of ChIP analysis conditions should also be clearly stated in the Methods section.

Response:

In the ChIP-seq experiments, *sigE* expressed from its native locus was HA-tagged. This σ^E -HA tag strain was then compared to a strain that was devoid of the tag as is good practice in the field. Therefore, we do not think that conducting the experiment also in a strain where the *sigE* gene is deleted is necessary/relevant.

However, we realized that this aspect (comparing σ^E -HA strain to a strain without tag) was not well described in the Mat & Met section.

Action taken:

We provided a more thorough description of the strains used in the ChIP-seq experiment in M&M section, chapter **Chromatin immunoprecipitation (ChIP)**. We added new **Supplementary Figure 2**, where you can see normalized genome coverage for ChIP-seq for wt (negative control) and σ^E -HA strain that were treated or not with EtOH.

2. The ChIP-Seq data analysing Sigma E and the gene expression pattern in Figure 3 must be validated under identical conditions. As shown in Figure 1, the effect of stress on the expression level of Sigma E was different, and the data were not from the same stress and the same timing, which makes it meaningless to consider it as an effect of Sigma E. In addition, the time scales in the sampling were completely different and therefore not comparable.

Response:

The fermentor study was used to find correlations between regulators and their targets. This was possible because the fermentor study was unprecedentedly detailed (32 time points) thereby allowing this type of analysis. The results (identified regulators-target genes) could then be applied also to other conditions. This was subsequently tested for selected genes (RT-qPCR), confirming validity of the approach.

Action taken:

We now mention the rationale behind using the expression dataset for kinetic modelling in the text (**page 7, line 146**).

3. Transcriptional regulation builds a network and hierarchical structure, with sigma factors and transcription factors regulating each other. Therefore, in order to discuss molecular mechanisms, the directness of gene expression regulation must be demonstrated, which is fatal to the lack of this study throughout.

Response:

The ChIP-seq data and kinetic analysis of gene expression were genome-wide and provided insights into gene expression regulation by σ^E . Selected examples were then tested and validated experimentally (RT-qPCR, promoter mutation analysis). Especially the direct negative effect of σ^E on expression of SCO4350 is strongly indicative, demonstrating a thus far overlooked role of σ factors. Of course, for some genes that were not tested experimentally, the regulation might be indirect but performing experiments with all those genes/promoters is out of scope of this manuscript. To conclude, we believe that the data strongly suggest a direct negative role of σ^E in gene expression. Future work may focus on other genes/mechanistic aspects of this phenomenon.

Action taken:

Effects of σ factors may be both direct/indirect. We now specify in the text that experimental evidence is required to distinguish between the two possibilities (**page 26, line 476**) for other genes than those tested. In the case of the tested genes, we moderated the language to indicate that future experiments will be required to fully understand the phenomenon.

4. The proposed molecular mechanisms are based only on the mRNA level, which is insufficiently validated and only speculative. For example, the models for Lines 221-229, Lines 230-242 and Lines 274-280 must be demonstrated to be direct effects and the possibility of indirect effects must be excluded.

Response:

- A. Lines 221-229: We agree, this is a hypothesis as we wished to provide some explanation for the observed effects. σ^R and BldN (two σ factors) emerged as two likely candidates. However, for our current study the important point was that σ^E functions as a direct repressor of SCO4350 (demonstrated by subsequent promoter mutation analysis) and the identity of the positive regulator of the gene was not that important. It can be addressed in a future study.
- B. Lines 230-242: We tried to provide an alternative explanation for the observed behavior of gene expression. We state that "we created a model", indicating it as a possibility and not certainty. Future experiments will be needed to test it.
- C. Lines 274-280: We agree, it is a hypothesis at this point. The direct/indirect effects were further validated (SCO7650, SCO4350).

Action taken:

- A. We rewrote this part and moved it into Discussion (**page 26, 486**).
- C. We address this in Discussion (see also **Response to your comment #3**): Effects of σ factors may be both direct/indirect. We now specify in the text that experimental evidence is required to distinguish between the two possibilities (**page 26, line 476**) for other genes than those tested.

5. Line 292-: states that mutations in the promoter sequence have resulted in Sigma E no longer binding, but this is speculative. It must be demonstrated that binding is actually altered. For example, it needs to be verified whether Sigma E and Sigma R are being replaced, as the authors would like to claim.

Response:

Yes, a mechanistic *in vitro* study could elucidate this aspect in more detail, and this may be a topic of a future study. Nevertheless, we believe that the mutation promoter assay results strongly suggest that σ^E , in the SCO4350 promoter region, is functioning as a direct repressor.

Action taken:

We moderated the claims across the manuscript. In Discussion, we mentioned that detailed mechanistic studies are needed for full understanding of the phenomenon (**page 26, line 478**).

6. The authors claim that the Sigma E model requires Sigma E binding by itself or Sigma E holo enzyme to be stopped by the promoter, but has this been demonstrated? The Sigma N holo enzyme of *E. coli* can act as a repressor as it stops the formation of a closed complex in the absence of an enhancer, but what about Sigma E in *S. coelicolor*? If there are findings, they should be cited.

Response:

We agree. Repression/activation can be modulated by other proteins. Using immunoprecipitation of σ^E , we detected some proteins (transcription factors) that can be involved. These are preliminary data, and the verification of specific interactions will be subject of future research. Nevertheless, we believe that this info will be of interest to the scientific field.

Action taken:

We added a sentence about a hypothetical enhancer acting on σ^E to Discussion (**page 27, line 501**).

Minor comments

7. Is the decrease in protein levels in Sigma E under NaCl, diamide and 42°C stresses in Fig. 1 a valid phenomenon? Relevant previous findings should be cited and explained.

Response:

We believe that it is indeed a valid phenomenon, the data indicate so! A similar phenomenon was detected in *e.g. Streptococcus mutans* where σ^X is rapidly degraded during escape from competence state (**PMID: 25005884**). It could also have a physiological relevance during specific stresses (see **response to comment 1 of Reviewer #3**).

Action taken:

We added information about σ factor degradation into the Discussion (**pages 23 and 25, lines 400 and 459 respectively**).

8. Why are SCO2970 and SCO6056 listed as Periplasmic/exported/lipoproteins in Functional Category in Table S3 when their Name is hypothetical protein? In other genes, hypothetical protein is an Unknown function.

Response:

Full citations of the database items for SCO2970 and SCO6056 are:

SCO	protein_name	Sanger_database_x00	Sanger_database_xx0	Sanger_database_xxx
2970	hypothetical protein	4.0.0 Cell envelope (0)	4.1.0 Periplasmic/exported/ lipoproteins (0)	4.1.6 Gram +ve membrane (836)
6056	hypothetical protein	4.0.0 Cell envelope (0)	4.1.0 Periplasmic/exported/ lipoproteins (0)	4.1.7 Gram +ve exported/lipoprotein (439)

We used _xx0 category for characterization of genes, where both of them are correctly assigned to the category Periplasmic/exported/lipoproteins.

9. It is difficult to understand the credibility of the target list; the binding strength (ratio?) of the ChIP should be shown in Tables S3-S6.

Response:

The fold enrichment and other MACS2 peak statistics is part of **Supplementary File 1**. It would be redundant to include them again in Tables S3-S6.

10. Similarly, consensus sequences and conservation (%) should also be indicated in Table S3-S6.

Response:

We agree.

Action taken:

We added the consensus sequences of identified motif sites into the legends of **Tables S3-6**, Conservation of the individual sites relative to each consensus were added to **Tables S3-6** as column named "*Identified Site Identity to Consensus (%)*".

11. Known and novel targets should be indicated in Table S3-S6.

Response:

This information actually was/is a part of **Supplementary file 4** as mentioned in the text (**page 24, line 423**)

12. Line 127: It would be easier to indicate previously identified sequences if they are also described and shown in the text.

Response:

We agree.

Action taken:

We added **Supplementary Figure S2B** showing a comparison of the logo identified by us with the previously identified logo.

13. The blank background in Table S4-S6 should be described as an indirect effect, as no Sigma E binding was observed.

Response:

The blank background indicates that those genes were found by ChIP-seq in the current study (and not in the study of Tran *et al.*) and they were found both in the presence/absence of ethanol. This was not explicitly stated in the legend.

Action taken:

We added the information about genes with blank background into the legends of **Tables S4-S6**.

14. The phase shift in Fig 3 does not distinguish between indirect and direct effects.

binding was observed.

Response:

There is no phase shift. If the question was meant as that there should be a phase shift between the σ factor profile and the regulated gene profile, then with respect to the time scale of the measured profile (hours) and the speed of expression (milliseconds), the expected time shift will be hidden in the experimental noise.

15. Line 153: With regard to the six sRNAs, how were they modelled (grouped) in the absence of kinetic data?

Response:

They were not modelled because there were no transcriptomic data for them. This is mentioned on **(page 7, line 154)**.

16. The timing and conditions of the expression analysis in Figure 4 are unclear.

Response:

Thank you for pointing this out.

Action taken:

We provided details about the expression analysis by RT-qPCR into the **M&M** section and mention this in the Figure legend.

As for the expression profiles (charts on the left) – these are the results of the computational modelling. This is now specified in the legend.

17. The effect of the presence or absence of EtOH addition should also be examined in Figure 7.

Response:

Yes, it is an interesting point. However, for the purposes of the manuscript we selected the non-EtOH conditions as most relevant according to RT-qPCR results, suggesting strong expression of SCO4350 in the absence of σ^E in normal conditions.

18. The consensus sequence in Figure 8 is not objective and cannot be discussed because the number of subjects is too small.

Response:

Yes, the number of subjects is small.

Action taken:

We included a note into the manuscript, stressing that the number of sequences on which it is based is small (**page 20, line 346**).

19. For objective evaluation in interaction experiments with Sigma E, gel images of the SDS-PAGE samples after pull-down should be shown.

Response:

Pull-down experiment was performed as it is described in M&M section. Relevant controls were used, and experiment was conducted in three biological replicates. We do not have SDS-gels after pull-down experiment. Primary data for each replica are accessible on PRIDE.

Action taken:

The pull-down experiment was described in more detail in **M&M** section.

20. As mentioned above, it has been demonstrated that Sigma N in E. coli acts as a repressor (DOI: 10.1099/mgen.0.000653). This reference must be cited in this paper as it is an example of RNAP holo working in repression.

Response:

We agree.

Action taken:

Information about σ^N and relevant references are now involved in Discussion (**Page 27, line 495**).

REVIEWERS' COMMENTS:

Reviewer #2 (Remarks to the Author):

No additional comments. Thank you for the thorough responses to my comments.

Reviewer #3 (Remarks to the Author):

The authors have satisfactorily answered point-by-point to my questions. The reviewer think that this manuscript reveal the opposite, inhibitory role of ECF sigma factor E protein by multiple molecular methods such as ChIP-seq in *Streptomyces coelicolor*. They also validated the findings of direct activation and repression function of sigma E in vivo, and suggested a common phenomenon among all the sigma factors and other organisms. The findings are important to expand our understanding of gene expression and regulation in bacteria. Now the manuscript could be accepted for publication in the current version.

Reviewer #4 (Remarks to the Author):

In the revised manuscript, the authors have appropriately responded to this reviewer's points, and the authors' arguments are clear and precise. It is unfortunate that the experiment does not demonstrate the direct mechanism of SigmaE, but the authors have weakened their claims to make it acceptable.

REVIEWERS' COMMENTS:

Reviewer #2 (Remarks to the Author):

No additional comments. Thank you for the thorough responses to my comments.

RESPONSE

Thank you!

Reviewer #3 (Remarks to the Author):

The authors have satisfactorily answered point-by-point to my questions. The reviewer think that this manuscript reveal the opposite, inhibitory role of ECF sigma factor E protein by multiple molecular methods such as ChIP-seq in *Streptomyces coelicolor*. They also validated the findings of direct activation and repression function of sigma E in vivo and suggested a common phenomenon among all the sigma factors and other organisms. The findings are important to expand our understanding of gene expression and regulation in bacteria. Now the manuscript could be accepted for publication in the current version.

RESPONSE

Thank you!

Reviewer #4 (Remarks to the Author):

In the revised manuscript, the authors have appropriately responded to this reviewer's points, and the authors' arguments are clear and precise. It is unfortunate that the experiment does not demonstrate the direct mechanism of SigmaE, but the authors have weakened their claims to make it acceptable.

RESPONSE

Thank you!